# Crystallization-Inspired Design and Modeling of Self-Assembly Lattice-Formation Swarm Robotics

**DOI:** 10.3390/s24103081

**Published:** 2024-05-12

**Authors:** Zebang Pan, Guilin Wen, Hanfeng Yin, Shan Yin, Zhao Tan

**Affiliations:** 1State Key Laboratory of Advanced Design and Manufacture for Vehicle Body, Hunan University, Changsha 410082, China; zbpan@hnu.edu.cn (Z.P.);; 2School of Mechanical Engineering, Yanshan University, Qinhuangdao 066004, China

**Keywords:** swarm robotics, self-assembly formation, crystallization, macroscopic model

## Abstract

Self-assembly formation is a key research topic for realizing practical applications in swarm robotics. Due to its inherent complexity, designing high-performance self-assembly formation strategies and proposing corresponding macroscopic models remain formidable challenges and present an open research frontier. Taking inspiration from crystallization, this paper introduces a distributed self-assembly formation strategy by defining *free*, *moving*, *growing*, and *solid* states for robots. Robots in these states can spontaneously organize into user-specified two-dimensional shape formations with lattice structures through local interactions and communications. To address the challenges posed by complex spatial structures in modeling a macroscopic model, this work introduces the structural features estimation method. Subsequently, a corresponding non-spatial macroscopic model is developed to predict and analyze the self-assembly behavior, employing the proposed estimation method and a stock and flow diagram. Real-robot experiments and simulations validate the flexibility, scalability, and high efficiency of the proposed self-assembly formation strategy. Moreover, extensive experimental and simulation results demonstrate the model’s accuracy in predicting the self-assembly process under different conditions. Model-based analysis indicates that the proposed self-assembly formation strategy can fully utilize the performance of individual robots and exhibits strong self-stability.

## 1. Introduction

The initial inspiration for swarm robotics derives from the incredible self-organizing swarms observed in nature, such as bird flocks [1], fish schools [2], and crystallization [3]. Such natural swarms formed by numerous simple units are completely distributed and decentralized, achieving complex collective behavior solely through simple local interactions rather than external or central controls [4]. This spontaneous phenomenon is defined as self-organization, and its characteristics, including scalability, flexibility, and robustness, serve as critical indicators for developing swarm robotics [5,6,7,8]. As a result, the focal point of the field of swarm robotics is to develop a swarm of simple robots. They can go beyond the capabilities of individual robots and effectively collaborate to achieve higher-level collective objectives [6]. Over the past two decades, swarm robotics has gradually evolved into a mature field, engaging researchers worldwide in its development. A comprehensive exploration of fundamental swarm behaviors, such as aggregation [9], foraging [10], and collective exploration [11], has been undertaken. The characteristics of these swarm behaviors are also analyzed by modeling microscopic [12,13] and macroscopic [14,15,16] models. Through these investigations, a more profound comprehension of swarm robotics and its prospective applications has been achieved.

Self-assembly is a reversible process in which pre-existing simple entities spontaneously form an ordered spatial structure without external intervention [17,18]. In the field of swarm robotics, self-assembly formation is considered a fundamental research topic and holds significant potential for practical applications [19]. These applications encompass the self-assembly control of the satellites [20,21], the programmable self-assembly matter [22,23], and advanced manufacturing rooted in self-assembly [24]. Consequently, this challenging and meaningful problem has attracted the attention of researchers worldwide striving to develop high-performance self-assembly swarm robotics. In 2014, Rubenstein et al. designed distributed local interaction rules for swarm robotics based on the finite-state machine (FSM) and achieved self-assembly formations by organizing a motion chain [25]. However, this approach operated in a single-threaded mode, resulting in extremely low efficiency. Following this, Yang et al. proposed a self-assembly formation based on a distributed algorithm by implementing two parallel motion chains [26,27]. Although this policy improved the efficiency, the high parallel potential of swarm robotics was not fully exploited. Additionally, Divband Soorati et al. designed swarm robotics to achieve the self-assembly of a tree formation [28]. Since father nodes constrained the growth of child nodes, the efficiency of the self-assembly was also hindered. In 2019, Zhu et al. defined four rules of collective behavior to form a square formation [29]. While this algorithm showed high parallelism and efficiency, its robustness was limited by the global leader. Zheng et al. [30] and Deshmukh et al. [31] discussed the self-assembly formation based on the density-feedback method, showing great scalability and high parallelism. However, the density-feedback laws relied on off-line global pattern planning [30] or a centralized controller [31], which conflicted with the flexibility and robustness of swarm robotics. The graph-based method was applied in a small-scale self-assembly by Klavins [32] and Mong-ying et al. [33]. Nevertheless, this control policy might face the challenge of dimensional explosion when applied to large-scale swarm robotics. Similarly, automated design methods were typically limited to small-scale self-assembly [34,35]. The potential field approach [36,37,38] might achieve distributed and high-efficient self-assembly formation. As the potential field only constrained robots to a specific area rather than certain positions, forming a formation with lattice structures was challenging. The above research on self-assembly of swarm robotics is summarized in Table 1. While the above research is valuable and offers essential guidance for exploring self-assembly formations, the potential of swarm robotics has yet to be realized. Indeed, crystallization is an exciting self-assembly phenomenon in nature, where the free particles efficiently and spontaneously form crystals with a periodic arrangement of atoms [39,40,41]. This phenomenon may contain the key principles for designing high-performance self-assembly formation strategies.

The macroscopic model provides a crucial theoretical basis for studying swarm robotics [4]. It can directly capture the critical features of swarm behavior and predict the long-term behaviors of swarm robotics [42,43]. Hence, macroscopic models facilitate the analysis of the underlying mechanisms driving swarm behavior [42,43]. These analysis results can be further used to optimize the design of the swarm robotics and improve the robot controllers. Generally speaking, macroscopic models can be subdivided into non-spatial models and spatial models [4,43]. The non-spatial model is widely used, relying on the assumption of spatial uniformity [44]. Based on the probabilistic finite-state machine (PFSM), Martinoli et al. [14] and Konur et al. [45] presented the non-spatial model for the collaboration-based stick-pulling swarm. The steady-state conditions and optimal collaboration rate were further discussed. Schmickl et al. [46] used the non-spatial model based on the stock and flow diagram to analyze the feedback loops of the aggregation behaviors. Liu et al. proposed a non-spatial model for foraging behaviors using PFSM [16]. On this basis, Song et al. found the optimal decision rules of the foraging behaviors [10]. The above research demonstrates that non-spatial models are usually very adept at analyzing the driving mechanism of swarm behavior besides the predicted ability. The spatial model is another significant macroscopic approach for studying swarm robotics, being mathematically grounded in the Fokker–Planck or diffusion-reaction equations [7,47,48]. Since the spatial model characterizes the ensemble of trajectories for a swarm within a specific area and time, its predictions can be visualized to directly depict the anticipated shape of the swarm robotics [4,7]. Schmickl et al. [46] and Prorok et al. [49] discussed spatial models in aggregation and coverage behaviors, respectively. They demonstrated little difference between spatial and non-spatial models in long-term predictions. Although the predictions of the non-spatial models are more visually intuitive, it is difficult to identify the driving mechanisms of swarm behaviors [46]. Meanwhile, due to the difficulty in obtaining analytical solutions, the applications of spatial models may be constrained [7].

Although macroscopic models have been applied extensively to study various swarm tasks and have shown great significance, modeling a macroscopic model for self-assembly formation remains a relatively underexplored area. Whether it is a spatial or non-spatial model, a key aspect of modeling is constructing the state transition functions (or state transition probabilities) between states [14,46]. To the authors’ knowledge, two widely used methods for constructing the state transition function are the data statistics method [50,51,52,53] and the geometrical estimation method [14,16]. However, in all current studies on modeling macroscopic models for self-assembly formation, the state transition functions are often reduced to experiment-based free parameters [14] using the data statistics method [50,51,52,53]. Since the free parameters fail to explain the inner mechanisms of the state transitions, the scientific value of the macroscopic model becomes limited. The state transition function, based on the geometrical estimation method, holds the potential to elucidate its transition mechanisms. However, this method is only suitable for situations where state transitions are only influenced by the geometric description of the robot’s sensing area and the density of the robots, e.g., stick-pulling [14], aggregation [46], and foraging [16]. In self-assembly formation, the state transition is also affected by the spatial structural features of the formation besides the robot’s sensing area. These complex spatial structures introduce strong nonlinearity to the state transitions. Therefore, the geometrical estimation method cannot be used to construct state transition functions for the self-assembly formation.

Designing a high-performance self-assembly formation strategy to achieve the two-dimensional (2D) shape formation with lattice structures and presenting a corresponding macroscopic model is challenging and meaningful for advancing swarm robotics. In this paper, a high-performance distributed self-assembly formation strategy is introduced, drawing inspiration from crystallization processes. To convert the macroscopic swarm behaviors to multiple simple collaborative tasks, five states, i.e., *free*, *moving*, *building*, *growing*, and *solid*, are defined for robots inspired by the crystallization’s phase transition processes (see Figure 1a and Table 2). Corresponding behavioral rules are designed for each robot state. Additionally, the concepts of unit cells and nucleation in crystallization are incorporated into self-assembly formation to initiate the self-assembly process and facilitate lattice formation, respectively. Consequently, robots in various states can autonomously collaborate and communicate locally, allowing them to form user-specified 2D shape formations with lattice structures. Notice that designing a self-assembly formation strategy for swarm robotics is not a thorough reproduction of the crystallization process due to the difference between robots and crystal. To implement the proposed self-assembly formation, Waxberry robots running within a grid ground and the embodied simulator are developed. Meanwhile, a non-spatial macroscopic model is proposed for the self-assembly formation. Here, the master equations of this model are created based on the stock and flow diagram. To solve the limitations of the geometric estimation and data statistics methods, the structural features estimation (SFE) method is proposed to construct the model’s state transition functions. Through defining the basic six structural types and analyzing their features, the state transition functions that have the potential to explain the transition mechanisms are constructed. Real-robot experiments and embodied simulations verify the proposed model’s predictive capability for self-assembly formation across various shapes, scales, and system parameters.

The main contributions of this paper are summarized in the following three aspects. Firstly, a high-performance distributed self-assembly formation strategy inspired by crystallizations is proposed for homogeneous swarm robotics. Since the design of a swarm-robotics cooperative mechanism in self-assembly formation is inspired by crystallization, swarm robotics shows excellent flexibility, scalability, and high efficiency. These advantages are demonstrated by real-robot experiments and simulations. Secondly, the SFE method is proposed to address the challenges posed by complex spatial structures in modeling macroscopic models. Compared with the data statistics method, the SFE method possesses the potential to delve into the mechanisms underlying swarm behaviors. Meanwhile, the proposed method overcomes the limitations of the geometric estimation method in scenarios with spatial structures. Besides the self-assembly formation, the proposed method is also significant in modeling other swarm tasks involving complex spatial structures. Thirdly, a non-spatial macroscopic model is modeled for the proposed self-assembled formation based on the stock and flow diagram and SFE method. Compared to simulations, the proposed model can quickly and accurately predict the whole macroscopic dynamic results. This predictive capability is an important guarantee for the realization of practical applications of self-assembly formation, especially in large scale tasks. Furthermore, the analysis based on the model indicates that shorter building time consumed by a single robot means a higher forming efficiency. It points out ways of optimizing the efficiency of the self-assembly formation.

The rest of this paper is organized as follows. In Section 2, the distributed self-assembly formation strategy inspired by crystallization is proposed and verified. In Section 3, the mathematical description of the macroscopic model is introduced. In Section 4, the validation of the proposed model and discussions are expounded. Lastly, the conclusions of this paper are drawn in Section 5.

## 2. The Self-Assembly Formation Inspired by Crystallization

### 2.1. Crystallization and Inspiration

Crystallization is a natural self-assembly process in which free solute particles (e.g., atoms, ions, or molecules) are arranged into highly ordered structures, resulting in a crystal with a periodic arrangement of atoms [39,40,41]. Driven by the laws of physics, free solutes spontaneously assemble into a crystal through local interactions without external controls. This phenomenon signifies a set of high-performance self-assembly rules. Consequently, it becomes feasible to devise a high-performance distributed self-assembly formation strategy for swarm robotics to create a 2D shape with lattice structures by emulating the crystallization process. Here, three critical characteristics in crystallization are worth noting, offering inspiration for self-assembly formation.

Firstly, crystallization is a phase transition [54]. Through diffusion (i.e., mass transfer), the free solute particles are transported from the solution to the growing crystal surface (see Figure 1a). According to the Kossel–Stranski model [55], the particles attached to the kinks are incorporated into the crystal by forming new bonds with crystal atoms on the surface, resulting in crystal growth. Indeed, the crystallization process can be regarded as cooperating task finished by the particles in different phases or states. This phenomenon suggests to us that the complex self-assembly formation can be decomposed into multiple simple collaborative tasks performed by robots in different states.

Secondly, the unit cell is the fundamental building block of the crystal and defines atoms’ basic arrangement and repetition [56]. It is the key to generating periodic latticed structures. Here, Figure 1b shows the definition of the unit cell according to Wigner–Seitz in the parallelogram lattices [56]. Therefore, introducing the unit cell for self-assembly is the key to achieving the formation with lattice structures. Under the above inspirations of the crystallization’s characteristics, the self-assembly formation strategy is introduced in Section 2.2.

Thirdly, crystallization begins in nucleation. Nucleation may occur spontaneously from the solvated phase or be artificially induced [57]. When the radius of nucleus r exceeds its critical radius, the crystal growth begins based on this stable nucleus (see Figure 1c). It is not hard to find that a stable point is essential for self-assembly.

The above characteristics can provide inspiration for designing the self-assembly formation of swarm robotics. However, due to the difference between crystals and robots, many unique crystallized mechanisms are not applied in the self-assembly formation of swarm robotics, such as the equilibrating crystallization process, the driving force of crystallization based on thermodynamics, and so on. Thus, designing the self-assembly formation strategy for swarm robotics is not a thorough reproduction and imitation of the crystallization process.

### 2.2. The Distributed Self-Assembly Formation Strategy for Swarm Robotics

For the self-assembly formation considered in this paper, all robots are initially distributed on the ground randomly and uniformly. This paper sets the density of robots on the ground as 0.2 by default. Generally, the number of robots is more than sufficient for self-assembly formation. Each robot has a low ability; they only know their position and can communicate with neighboring robots in a small local area. Note that direct communication has the advantages of portability and has a lower cost than stigmergy [58]. After giving the user-specified shape, robots will assemble into this formation by interacting with neighbors without relying on external control and global information. The details are designed as follows.

As introduced in Section 2.1, the inspiration drawn from crystal growth provides a framework for designing self-assembly formation. To decompose the complex self-assembly formation task into several simple cooperative tasks, five states are defined for robots inspired by the crystallization’s phase transition (see Table 2). Inspired by the free solute particles, the *free* robot is designed to provide the available materials for self-assembly formation. Like solute molecules in the diffusion process, the *moving* robot is responsible for transporting itself to the growth boundary of the formation. The particles attached to the kinks inspire the design of the *building* robot, which occupies a specified position to enter and expand the formation. The *growing* and *solid* robots correspond to atoms on the surface of and inside the crystal, respectively. Considering the differences between crystallization and self-assembly formation, the *growing* robot should provide the specified positions and recruit *free* and *moving* robots to these positions, promoting the expansion of formation. The *solid* robots no longer do anything, like the atoms inside the crystal. In addition, more sub-states are also defined for robots based on different criteria, out of consideration for modeling the macroscopic model. See Figure 2 and Section 3 for more details. As shown in Figure 1b, based on the Wigner–Seitz unit cell in the parallelogram lattices, the cross-arranged unit cells are designed for self-assembly formation to achieve the shape with lattice structures. Here, the unit cell is defined as a square grid with two unit lengths, allowing a *building*, *growing*, or *solid* robot to be positioned at its center (see Figure 3). The cross-arranged unit cells stipulate the arrangement of the robots in the formation and provide a basis for the *growing* robot to specify where it needs to be built. Referring to the definition of the cross-arranged unit cells (see Figure 3), each *growing* robot positioned in a unit cell has four neighboring unit cells. Within this context, the *growing* state is subdivided into three distinct sub-states, i.e., *G1*, *G2*, and *G3*, based on the conditions of neighboring unit cells (see Figure 2). As shown in Figure 4, the *G1*, *G2*, and *G3* states indicate that this *growing* robot has one, two, and three neighboring filled unit cells, respectively. The unit cell filled by *growing* or *solid* robots is defined as the filled unit cell; otherwise, it is the empty unit cell. These sub-states of the *growing* state are necessary for modeling the macroscopic model of self-assembly formation. For more details, see Section 3. In addition, inspired by the nucleation in crystallization, two robots will be selected as the initial *growing* robots and positioned at the coordinates $\left(x_{0},\;y_{0}\right)$ and $\left(x_{0},\;y_{0}+2\right)$ according to the cross-arranged unit cell’s definition (see Figure 5). $x_{0}$ and $y_{0}$ can be assigned any value. The configuration of two *growing* robots can trigger self-assembly and benefit subsequent modeling macroscopic models. Other robots will be configured as *free* robots and placed in the arena randomly and uniformly, similar to solute particles (see Figure 5). Then, the self-assembly formation will start automatically. The detailed control strategy of the robots in different states is introduced as follows.

When a *growing* robot is generated within a unit cell, its four neighboring unit cells that share an edge with its unit cell will be defined simultaneously (see Figure 3). Unit cells not yet filled by *growing* or *solid* robots are defined as empty; otherwise, they are defined as filled. Table 3 introduces the classifications and definitions of unit cells. Therefore, like the atoms on a growing crystal surface, the *growing* robots can recruit others to occupy their neighboring empty unit cells to expand the formation. Specifically, the *growing* robot selects one of the neighboring empty unit cells as an available empty unit cell. The selected one should be closest to the coordinate origin and not occupied by *building* robots. The coordinates of the available empty unit cell’s center are then broadcast to recruit *free* robots. As shown in Figure 6a, when the *growing* robot first broadcasts its selected coordinates at time step t, this broadcast only impacts the *free* robots. After receiving recruitments, the *free* robot turns into the *moving* state and selects the nearest available empty unit cell as its target position, moving towards it. During the move, the *moving* robot will not actively change its target position. To avoid self-assembly downtime, the moving time parameter $T_{M}$ is defined. Thus, *moving* robots may turn into *free* robots due to exceeding the moving time parameter $T_{M}$. Furthermore, as the *growing* robots’ broadcasts are public, multiple *moving* robots may hold the same target position. Hence, the competitive mechanism is applied based on the distance to the target position and ID number. As shown in Figure 6b, *moving* robots numbered 1, 3, and 6 share the same target position. The *moving* robot closest to the target position with the lowest ID number value can win the competition. Thus, in Figure 6b, the *moving* robots numbered 3 and 6 fail in the competition and will degenerate into *free* robots. The *moving* robot numbered 1 wins the competition and moves towards its target position. The *moving* robot that arrives at its target position will translate into a *building* robot (see Figure 6c). The *building* robots located in unit cell will wait for a few time steps (i.e., building time parameter $T_{B}$) to simulate the time-consuming of assembly and building work. Whereafter, they will become the *growing* robots, resulting in the expansion of the formation. If the currently selected available empty unit cell is occupied by a *building* robot, the *growing* robot will continue to choose a new available empty unit cell and broadcast for it. When all neighboring unit cells are filled unit cells, this robot completes its task and transitions from the *growing* state into the *solid* state. As shown in Figure 7, the finite-state machine (FSM) shows the whole control flow of the individual robot in the proposed self-assembly formation.

### 2.3. The Implementation of the Self-Assembly Formation

As shown in Figure 8 and Figure 9, the Waxberry robots are designed to achieve the proposed self-assembly formation, operating on a grid ground composed of unit square grids. The name “Waxberry” is derived from a unique fruit in China. As shown in Figure 8b, each Waxberry robot is equipped with two independent stepping motors. Under the control of the motor driver, each robot wheel can independently rotate forward or reverse at a specified speed. Thus, the Waxberry robot can move omnidirectionally based on the differential steering control. Here, the unit length and unit area are defined as the side length and area of a square unit grid, respectively. On the grid ground, the Waxberry robot can navigate between the locating points positioned on the nodes of each square unit grid (see Figure 9). Therefore, after obtaining its initial coordinates, the Waxberry robot can accurately determine its real-time position by tracking its trajectory using the principles of inertial navigation [59]. For simplicity, each movement from one locating point to any of its eight adjacent points is referred to as one time step. The local communication range of the Waxberry robot is limited to the area that the robot can reach in three time steps (see Figure 9). The information exchanged among Waxberry robots includes the robot’s ID number, state, present coordinates, and working information.

In this study, 15 Waxberry robots are manufactured. Each Waxberry robot is equipped with an identical self-assembly strategy, as described in Section 2.2. Two different shape formations are considered to test the proposed self-assembly strategy, i.e., the four-pointed star and the hexagonal crystal shape formations (see Figure 10). The system parameters of the self-assembly formation, i.e., building time parameter $T_{B}$ and moving time parameter $T_{M}$, are set as one step and four steps by default. As shown in Figure 10, under the control of the proposed self-assembly formation strategy, 15 Waxberry robots autonomously assemble into 2 different shape formations without relying on external controls and the global leader. To verify the proposed self-assembly formation in a large-scale swarm, an embodied simulator capable of accommodating thousands of agents is developed. This simulator is programmed in MATLAB 2020, a high-level programming language designed for engineers and scientists. Using MATLAB’s powerful development capabilities, the simulator allows a large number of agents to simultaneously emulate the functionalities of the Waxberry robot. Each agent in the simulator can move, communicate, and make autonomous decisions. Consequently, the proposed self-assembly can be further tested at large-scale formations. Here, the formation scale is expanded to include about 300 robots, 1500 robots, and 3500 robots, respectively. Based on the proposed self-assembly strategy, swarm robotics can successfully form the two considered shape formations. Figure 11 shows the self-assembly process when the formation is composed of about 300 robots.

The above real-robot experiments and simulations demonstrate the proposed self-assembly strategy’s high flexibility, scalability, and efficiency. The robots in different states can self-assemble into various shape formations through local interactions and cooperation, demonstrating flexibility in different shape conditions. Meanwhile, compared to the works in Refs. [29,30,31], the proposed distributed self-assembly formation does not rely on a global leader and pre-design global information. This allows swarm robotics more flexible deployment capabilities, showing massive potential in practical applications. Additionally, tests conducted at the formation scale, ranging from about 15 to 3500 robots, verify that the proposed self-assembly strategy exhibits excellent scalability. Such scalability comes from a fully distributed self-assembly strategy based on local interactions. That is, the local environments of individual robots are similar, no matter how large the scale of the swarm robotics. High forming efficiency is also one of the advantages of the proposed self-assembly formation. Compared with the one or two parallel motion chains in Refs. [25,26,27], the proposed self-assembly formation shows remarkable parallelism capability, as all *growing* robots contribute to the expansion of the formation. Indeed, as the formation grows larger, the forming efficiency becomes higher due as there are more *growing* robots involved in the construction. Thus, the proposed self-assembly formation shows a rare and precious super-linear feature. This feature is analyzed in detail below.

The super-linear feature is strong evidence for the high forming efficiency. That is, the forming efficiency of the self-assembly increases with the scale of the formation [13]. It is a rare and precious feature. The efficiency of the existing self-assembly formation designed by Rubenstein et al. [25] and Yang et al. [26] remains constant and does not increase as the formation scale becomes larger (see Figure 12). To analyze the super-linear feature, the average number of robots joining the formation within one time step is defined as the forming efficiency index $e_{c}$. It can be expressed as follows.
(1)$$e_{c}=\frac{m_{0}}{T}$$
here, T represents the total time dedicated to the self-assembly formation, while $m_{0}$ denotes the number of robots positioned within the formation (i.e., the formation scale). As shown in Figure 12, whether forming the four-pointed star shape or the hexagonal crystal shape formation, the simulation results show that the forming efficiency increases with the scale of the formation. In the small-scale formation comprising about 15 robots, the efficiencies $e_{c}$ of forming the four-pointed star shape formation and the hexagonal crystal shape formation are 0.764 and 0.598, respectively. However, when the scale formation is expanded to about 3500 robots, the efficiencies $e_{c}$ are 13.418 and 14.455, respectively. The efficiencies of forming the four-pointed star shape and the hexagonal crystal shape formations are increased by 1756% and 2417%, as the formation scale increases from about 15 to 3500 robots. The efficiencies $e_{c}$ of self-assembly formation designed by Rubenstein et al. [25] and Yang et al. [26] can be viewed as one and two, respectively, using robots of the same performance without considering the preparation time. Although our proposed self-assembly formation cannot show an efficiency advantage over the work in Refs. [25,26], the efficiency of the proposed self-assembly formation is greater than 2 when more than 100 robots form the formation (see Figure 12). In the formation comprising about 3500 robots, the efficiency of the proposed self-assembly formation is 13.9 and 6.97 times more efficient than the studies of Rubenstein et al. [25] and Yang et al. [26], respectively.

## 3. Mathematical Description of the Macroscopic Model

The macroscopic model is an important mathematical tool for predicting and studying swarm behaviors. This section develops a non-spatial macroscopic model for the proposed distributed self-assembly formation. The master difference equations (i.e., the model’s framework) are described in Section 3.1. The analysis for the structural features of the shape formation is introduced in Section 3.2. Subsequently, all state transition functions in this macroscopic model are estimated in Section 3.3 according to the structural features. Lastly, the effects of the shape boundary’s constraints are shown in Section 3.4.

### 3.1. The Master Difference Equations

In this section, the non-spatial macroscopic model is developed to study the characteristics of the proposed self-assembly formation’s swarm behaviors. Referring to Ref. [46], a stock and flow diagram (see Figure 13) is created to depict the swarm behaviors of the proposed self-assembly formation, utilizing the FSM depicted in Figure 7. The stocks, represented by boxes in Figure 13, denote the average number of robots in various predefined states at the macroscopic level. Arrows in Figure 13 depict flows, expressing the number of robots transitioning between states. Such changes can be described by state transition functions. Based on this stock and flow diagram, a set of difference equations in the discrete-time domain can be derived to model the fluctuations in the average number of robots between different states. The framework of the macroscopic model is outlined as follows.
(2)$$N_{F}\left(t+1\right)=N_{F}\left(t\right)-\Gamma_{F}\left(t\right)+\Gamma_{M}^{F}\left(t\right)+\Lambda_{M}^{F}(t,T_{M})$$
(3)$$N_{M}\left(t+1\right)=N_{M}\left(t\right)-\Gamma_{M}^{F}\left(t\right)-\Gamma_{M}^{B}\left(t\right)-\Lambda_{M}^{F}(t,T_{M})+\Gamma_{F}\left(t\right)$$
(4)$$N_{B}\left(t+1\right)=N_{B}\left(t\right)-\Lambda_{B}^{G}(t,T_{B})+\Gamma_{M}^{B}\left(t\right)$$
(5)$$N_{G1}\left(t+1\right)=N_{G1}\left(t\right)+\Lambda_{B}^{G1}\left(t,T_{B}\right)-{∆}_{G1}^{G2}(t)-{∆}_{G1}^{G3}(t)-\Omega_{G1}(t)$$
(6)$$N_{G2}\left(t+1\right)=N_{G2}\left(t\right)+\Lambda_{B}^{G2}(t,T_{B})+{∆}_{G1}^{G2}(t)-{∆}_{G2}^{G3}(t)-\Omega_{G2}(t)$$
(7)$$N_{G3}\left(t+1\right)=N_{G3}\left(t\right)+{∆}_{G1}^{G3}(t)+{∆}_{G2}^{G3}(t)-{∆}_{G3}^{S}(t)$$
(8)$$N_{S}\left(t+1\right)=N_{S}\left(t\right)+{∆}_{G3}^{S}\left(t\right)+\Omega_{G1}\left(t\right)+\Omega_{G2}\left(t\right)$$

As shown in Table 4, $N_{F}\left(t\right)$, $N_{M}\left(t\right)$, $N_{B}\left(t\right)$, and $N_{S}\left(t\right)$ are the average number of robots in state *free*, *moving*, *building*, and *solid* respectively at time step t. $N_{G1}\left(t\right)$, $N_{G2}\left(t\right)$, and $N_{G3}\left(t\right)$ represent the average number of robots in state G1, G2, and G3 at time step t. The definitions of the G1, G2, and G3 states can be found in Section 2.2 and Figure 4. Furthermore, the total number of *growing* robots is expressed as $N_{G}\left(t\right)$.

The state transition functions describe the changes in the number of robots transitioning between different states at time step t. The details are introduced as follows. $\Gamma_{F}\left(t\right)$ represents the number of *free* robots that turn into the *moving* state upon receiving broadcasts from *growing* robots. $\Gamma_{M}^{F}\left(t\right)$ represents the number of robots in the *moving* state that degenerate into the *free* state due to failure in competition with other *moving* robots for the same target position. $\Gamma_{M}^{B}\left(t\right)$ represents the number of *moving* robots that succeed in competition and arrive at their target positions. $\Lambda_{M}^{F}(t,T_{M})$ denotes the number of *moving* robots that run out of the moving time parameter $T_{M}$ and become the *free* state. $\Lambda_{B}^{G1}\left(t,T_{B}\right)$ and $\Lambda_{B}^{G2}\left(t,T_{B}\right)$ represent the number of *building* robots that transition into G1 and G2 states, respectively, after spending $T_{B}$ steps assembling the formation. Their total number is expressed as $\Lambda_{B}^{G}(t,T_{B})$. When the *building* robots occupying the empty unit cells become the *growing* robots, the existing *growing* robots will update their sub-states or translate into the *solid* state. These transitions are denoted as ${∆}_{G1}^{G2}(t)$, ${∆}_{G1}^{G3}(t)$, ${∆}_{G2}^{G3}(t)$, and ${∆}_{G3}^{S}(t)$, respectively. Additionally, $\Omega_{G1}(t)$ and $\Omega_{G2}(t)$ signify the count of *G1* and *G2* robots directly transitioning into *solid* robots due to due to the constraints of the shape boundary, respectively. At this point, the framework of the proposed macroscopic model is described completely.

### 3.2. The Analysis for the Structural Features

In the proposed self-assembly formation, all *growing* robots collectively constitute the growth boundary of the formation. The arrangement of *growing* robots based on the cross-arranged unit cells inevitably creates intricate structures within the growth boundary, causing strong nonlinearity for the state transitions. As shown in Table 5, the influence of spatial structures renders traditional geometrical estimation methods ineffective in deducing corresponding state transition functions. Similarly, the data statistics method falls short in elucidating the internal mechanisms of state transitions. Consequently, analyzing the structural features within the growth boundary of the formation is an essential precondition for constructing state transition functions capable of probing into the mechanisms of swarm behaviors. Although the macroscopic model considers the features of spatial structures, it remains non-spatial as the probability of these structural features affecting each robot is uniform. Here, the definition and quantities of six basic structural types are discussed to depict the structural features of the growth boundary. Subsequently, the distribution characteristics of *building* robots within the empty unit cells provided by different structural types are discussed. The arrangement characteristics of the structural types are also analyzed. Note that the distribution characteristics of *building* robots in empty unit cells and the arrangement characteristics of structural types are regarded as independent in probability. Additionally, some assumptions for the macroscopic model should be elucidated to facilitate the discussion of structural features. As introduced in Section 2.2, all robots share the same ability and an identical self-assembly strategy. In the task of the self-assembly formation, all robots distributed across the arena randomly and uniformly will experience *free*, *moving*, *building*, *growing*, and *solid* states and perform corresponding tasks to complete the self-assembly. Therefore, based on the principle of equalitarianism and the actual situations of self-assembly formation, the macroscopic model should satisfy the following assumptions:

**Assumption 1.** 
*The building efficiency of any unit cell remains constant both in terms of time and space.*


**Assumption 2.** 
*The formation expands at a consistent rate in all directions.*


**Assumptions 1** and **2** are consistent with the fact that the model is non-spatial. In the structural features analysis, these assumptions can reduce the number of situations that need to be discussed (see Section 3.2) and simplify the complexity of the model. Note that the above assumptions are only for the self-assembly of swarm robotics and have no bearing on crystallization.

#### 3.2.1. The Definitions of Six Basic Structural Types

As shown in Figure 14, six basic structural types are defined based on *G2* robots. Each structural type comprises only one *G2* robot, positioned on the leftmost side of the structural type. Thus, the total count of structural types corresponds to the number of *G2* robots. The structural types consisting of only *G2* and *G3* robots can be classified as Fa and Ra based on whether the angle between *G2* and its neighboring *growing* robot to the right is 180° (flat angle) or 90° (right angle) (see Figure 14a,c). When the structural type Fa contains a *G3* robot, it is subdivided into ${Fa}_{2}$; otherwise, it is ${Fa}_{1}$. Similarly, the structural type Ra is also divided into ${Ra}_{1}$ and ${Ra}_{2}$. The situations involving structural types that include more than one *G3* robot are not considered in this paper since the *G3* robot builds its neighboring empty unit cells with the same efficiency as other *growing* robots (see Assumption 1). To simplify the analysis, *G1* robots are only considered within structural types Fa, i.e., structural types ${Fa}_{1}$ and ${Fa}_{2}$ (see Figure 14e,f). That is, we ignored some of the rare structure types without reducing the accuracy of the model predictions. Using these defined six basic structural types, the growth boundary’s structural feature can be described precisely. The notations related to all state transition functions are explained in Table 6.

#### 3.2.2. The Calculation of the Quantities of Six Basic Structural Types

According to **Assumptions 1** and **2**, the shape of the formation’s growth boundary must be a convex polygon before the shape boundary constraint is triggered. As shown in Figure 15, the structural types Fa can bend the extension of the growth boundary by 90° in addition to increasing the length of the growth boundary. Therefore, based on the basic geometric principles, a growth boundary allows for only four structural types of Fa. The growth boundary can be understood as composed of four equal-length growth edges originating from the structural type Fa. The simplest growth edge contains only a structural-type Fa (see Figure 15a). Thus, the number of structural-type Fa is written as $N_{Fa}(t)$, i.e.,
(9)$$N_{Fa}(t)\;=min\left(4,N_{G2}\left(t\right)\right)$$
where $N_{G2}\left(t\right)$ expresses the number of *G2* robots. The number of structural types of ${Fa}_{1}$, ${Fa}_{1^{*}}$, ${Fa}_{2}$, and ${Fa}_{2^{*}}$ at time step t can be expressed as $N_{{Fa}_{1}}\left(t\right)$, $N_{{Fa}_{1^{*}}}\left(t\right)$, $N_{{Fa}_{2}}\left(t\right)$, and $N_{{Fa}_{2^{*}}}\left(t\right)$, i.e.,
(10)$$N_{{Fa}_{1}}\left(t\right)=N_{Fa}(t)(1-\rho_{G3})(1-\rho_{G1})$$
(11)$$N_{{Fa}_{1^{*}}}\left(t\right)=N_{Fa}(t)(1-\rho_{G3})\rho_{G1}$$
(12)$$N_{{Fa}_{2}}\left(t\right)=N_{Fa}(t)\rho_{G3}(1-\rho_{G1})$$
(13)$$N_{{Fa}_{2^{*}}}\left(t\right)=N_{Fa}(t)\rho_{G3}\rho_{G1}$$
where $\rho_{G1}$ is the probability of *G1* robots in structural type Fa, which is defined as
(14)$$\rho_{G1}=\frac{N_{G1}\left(t\right)}{N_{Fa}(t)}$$

The $\rho_{G3}$ denotes the probability that structural types Fa and Ra contain a *G3* robot. Due to equalitarianism, $\rho_{G3}$ can be expressed as
(15)$$\rho_{G3}=\frac{N_{G3}\left(t\right)}{N_{G2}\left(t\right)}$$

Here, $N_{G2}\left(t\right)$ represents the total number of all structural types. The number of the structural type Ra can be written as $N_{Ra}(t)$, i.e.,
(16)$$N_{Ra}(t)=N_{G2}\left(t\right)-N_{Fa}(t)$$

Similarly, the number of structural types ${Ra}_{1}$ and ${Ra}_{2}$ is represented as follows.
(17)$$N_{{Ra}_{1}}(t)=N_{Ra}(t)\;(1-\rho_{G3})$$
(18)$$N_{{Ra}_{2}}(t)=N_{Ra}(t)\;\rho_{G3}$$

#### 3.2.3. The Distribution Characteristics of Building Robots

The distribution of *building* robots within the empty unit cells provided by the structural types should be discussed since it plays a crucial role in the state transitions of *building* and *growing* robots. The empty unit cells allowed to be occupied by the *building* robots are considered the building unit cells (see Table 3 and Figure 14). Therefore, the distribution rules and classification of the building unit cells are determined first. Then, the sub-states of *building* robots should be discussed further. Finally, the probability of different *building* robots in different building unit cells can be determined.

As a prerequisite for estimating the number of building unit cells, based on a large number of simulations and experimental results, the following two phenomena related to empty unit cells deserve attention. Firstly, the building time parameter of *building* robots considered in this work is very small (i.e., $T_{B}\leq 4$), usually less than the time required for the robot to transition from *free* to *building* state. Therefore, in the case of a *growing* robot with multiple empty unit cells (such as *G1* and *G2* robots), when it autonomously recruits a *building* robot for the second empty unit cell, the *building* robot within its first empty unit cell has already transitioned to the *growing* state due to completing its assembly works. As this transition also triggers the original *growing* robot to undergo a state transfer, it is impossible for a *growing* robot to actively recruit multiple *building* robots simultaneously without triggering its state transition. Secondly, the empty unit cell shared by two neighboring *growing* robots is typically closer to the coordinate origin than its neighbor. According to the self-assembly strategy, *growing* robots prioritize selecting empty unit cells of this kind and broadcast their coordinates to recruit *free* and *moving* robots. Consequently, empty unit cells shared by two adjacent *growing* robots are more likely to be occupied by *building* robots, thereby becoming the building unit cells. Under the premise of satisfying the above phenomena, the simplified distribution rules for building unit cells are defined as follows to minimize model complexity.

(I)Each *growing* robot is deemed to actively provide only one empty unit cell as the building unit cell.(II)The *G1* robot must share the building unit cell with the *G2* robot in the current structural type.(III)The building unit cells actively provided by the *G2* robot should be unaffected by the structural types on the left.(IV)*G2* robots preferentially select empty unit cells shared with adjacent *growing* robots of the same structural types as the building unit cell.

Here, the total number of building unit cells can be written as follows.
(19)$$N_{bu}\left(t\right)=N_{{Fa}_{1}}\left(t\right)+2N_{{Fa}_{2}}\left(t\right)+N_{{Fa}_{1^{*}}}\left(t\right)+2N_{{Fa}_{2^{*}}}\left(t\right)+N_{{Ra}_{1}}\left(t\right)+N_{{Ra}_{2}}\left(t\right)$$

The green dotted boxes in Figure 14 represent the possible building unit cells. Note that a *building* robot cannot become a *G3* robot directly under current situations.

Indeed, a particular type of building unit cell, i.e., a priority building unit cell, is given preference for occupation by the *building* robots. According to the self-assembly strategy, once a *building* robot occupies a neighboring empty unit cell, the *growing* robot will immediately broadcast the next suitable neighboring unit cell’s coordinates. It is not hard to see that the corresponding building unit cell’s coordinates have been broadcast before a *growing* robot is upgraded to the *G3* sub-state. Consequently, the building unit cells provided by *G3* robots are designated priority building unit cells. In structural types ${Fa}_{1^{*}}$ and ${Fa}_{2^{*}}$, the simultaneous birth of two neighboring *G1* and *G2* robots from the *building* state is typically not feasible. Thus, when a *G1* robot sits next to a *G2* robot, their sharing building unit cell has been broadcast and should be treated as the priority. Besides the proposed self-assembly strategy, the generation of priority building unit cells is also related to the autonomous actions of robots. Estimating the priority building unit cell’s number based only on the self-assembly strategy and structural features is prone to bias. Considering the randomness of autonomous actions of robots, it is assumed that there are ξ priority building unit cells exiting in structural types ${Ra}_{1}$ to correct the estimate of the priority building unit cell’s number. Note that the autonomous actions of robots are affected by the external environment, i.e., the proportion of robots in different states within the swarm. As defined in Section 2.2, the building time parameter $T_{B}$ can change the survival duration of the *building* and *growing* robots to change the proportion of robots in different states within the swarm. Thus, based on the statistical result of simulations, Table 7 shows the values of ξ under different building time parameters $T_{B}$. The total number of the priority building unit cells is written as follows.
(20)$$N_{bu_{p}}\left(t\right)=N_{{Fa}_{1^{*}}}\left(t\right)+2N_{{Fa}_{2^{*}}}\left(t\right)+N_{{Fa}_{2}}\left(t\right)+N_{{Ra}_{2}}\left(t\right)+\xi$$

Other building unit cells are considered non-priority building unit cells, except for priority building unit cells.

Additionally, it is imperative to delve deeper into the sub-states of the *building* robot. These sub-states can be used to distinguish the neighboring *growing* robot’s working modes. This differentiation is crucial for constructing the state transition functions $\Gamma_{M}^{F}\left(t\right)$. Here, the *building* robots transitioning from the *moving* robot at time step t−1 are referred to as the *new building* robots, while others are classified as *old building* robots (see Figure 2b). According to **Assumption 1** and the discussion of the priority building unit cells, *old building* robots are given preference for occupying the priority building unit cells compared to *new building* robots. Thus, the probability of *old building* robots being distributed in the priority building unit cell is
(21)$$\sigma_{old}=min(1,\frac{N_{B_{old}}\left(t\right)}{N_{bu_{p}}\left(t\right)+e_{0}})$$
where $e_{0}$ is a value of $10^{-5}$ to avoid miscalculation. The $N_{B_{old}}\left(t\right)$ expresses the number of *old building* robots at time step t, which can be written as follows.
(22)$$N_{B_{old}}\left(t\right)=N_{B}\left(t\right)-N_{B_{new}}\left(t\right)$$
where $N_{B_{new}}\left(t\right)$ is the number of *new building* robots at time step t, which is equal to $\Gamma_{M}^{B}\left(t-1\right)$ (see Equation (3)). The probability that the *old building* robots are distributed in the non-priority building unit cell is
(23)$$\epsilon_{old}=\frac{N_{B_{old}}\left(t\right)-\sigma_{old}N_{bu_{p}}\left(t\right)}{N_{bu}\left(t\right)-N_{{bu}_{p}}\left(t\right)+e_{0}}$$

The probabilities of the *new building* robot distributed in the priority building unit cell and the non-priority building unit cell are
(24)$$\sigma_{new}=\min\;(1-\sigma_{old},\frac{N_{B_{new}}\left(t\right)}{N_{{bu}_{p}}\left(t\right)+e_{0}})$$
(25)$$\epsilon_{new}=\frac{N_{B_{new}}\left(t\right)-\sigma_{new}N_{bu_{p}}\left(t\right)}{N_{bu}\left(t\right)-N_{{bu}_{p}}\left(t\right)+e_{0}}$$

#### 3.2.4. The Arrangement Characteristics of Six Basic Structural Types

The arrangement of these structural types is another key factor affecting the state transitions. Note that the arrangement characteristics of the structural types and the distribution characteristics of the *building* robots are probabilistically independent of each other. According to the analysis of *building* robot distributions within building unit cells, the structural type ${Ra}_{1}$ plays a unique role. As shown in Figure 16, when a *growing* robot is born in the building unit cell provided by the structural type ${Ra}_{1}$, both the *G2* robot in the current structural type ${Ra}_{1}$ and another *G2* robot in the right structural type will update to *G3* robots simultaneously. That is, the structural type ${Ra}_{1}$ can affect its adjacent right structural types. This is a special feature that other structural types don’t have. Therefore, the key of the arrangement analysis is to estimate the probability that the left side of the focused structural type is structural type ${Ra}_{1}$. Here, based on the principle of equalitarianism, all structural types ${Ra}_{1}$ are evenly allocated to four growth edges and are arranged after the starting structural type Fa. Then, the structural types ${Ra}_{2}$ have an equal probability of appearing in all vacancies generated by structural types Fa and ${Ra}_{1}$ (see Figure 17). Utilizing permutation and combination theory [60], we can calculate the probability of structural type ${Ra}_{1}$ being on the left side of structural types ${Ra}_{2}$ and Fa as follows.
(26)$$\tau =\min\;(1,\frac{N_{{Ra}_{1}}\left(t\right)}{N_{Fa}\left(t\right)+N_{{Ra}_{2}}\left(t\right)})$$
where $N_{Fa}\left(t\right)+N_{{Ra}_{2}}\left(t\right)$ represents total number of the vacancies generated by structural types ${Ra}_{2}$ and Fa. The probability that a structural type ${Ra}_{1}$ is on the left side of another structural type ${Ra}_{1}$ is
(27)$$\tau^{*}=\frac{\max\;(0,N_{{Ra}_{1}}\left(t\right)-N_{Fa}\left(t\right)-N_{{Ra}_{2}}\left(t\right))}{N_{Fa}\left(t\right)+N_{{Ra}_{2}}\left(t\right)}$$

### 3.3. The State Transition Functions Based on the SFE

#### 3.3.1. The State Transition Function $\Gamma_{F}\left(t\right)$

The $\Gamma_{F}\left(t\right)$ represents the number of *free* robots transitioning to the *moving* state due to receiving recruitments of the *growing* robots. It is written as follows.
(28)$$\Gamma_{F}\left(t\right)=S_{0}d_{0}-N_{M}\left(t\right)-N_{B}\left(t\right)\;$$
where $S_{0}$ represents the total communication area created by all *growing* robots. Note that the communication range of a *growing* robot can also cover the formed formation. This area will not be taken into consideration for $S_{0}$ (see Figure 18). Within this communication area, $d_{0}$ denotes the robot’s density, which is defaulted to 0.2 (see Section 2.2). $S_{0}d_{0}$ represents the total number of robots. Furthermore, all *moving* and *building* robots are situated within the total communication area, enabling interaction with *growing* robots. Therefore, $\Gamma_{F}\left(t\right)$ can be expressed by Equation (28).

The total communication area $S_{0}$ created by all *growing* robots is the key to constructing the state transition function $\Gamma_{F}\left(t\right)$. As shown in Figure 9, the communication range of a robot can be regarded as 7×7 position points. Due to the defined cross-arrangement, there will be some overlap in the communication ranges of two neighboring *growing* robots (see Figure 19). To avoid repetitive computation, Equation (29) is proposed to calculate the total communication area based on the structural types. It can be written as follows.
(29)$$S_{0}=\alpha_{1}N_{{Fa}_{1}}\left(t\right)+\alpha_{2}N_{{Fa}_{1^{*}}}\left(t\right)+\alpha_{3}N_{{Fa}_{2}}\left(t\right)+\alpha_{4}N_{{Fa}_{2^{*}}}\left(t\right)+\alpha_{5}N_{{Ra}_{1}}\left(t\right)+\alpha_{6}N_{{Ra}_{2}}\left(t\right)-\alpha_{0}D_{G2}$$

To calculate the coefficients $\alpha_{0}$ to $\alpha_{6}$, the independent communication ranges of *growing* robots that do not overlap are defined artificially. Solid dots in Figure 18 and Figure 19 denote a unit area. As shown in Figure 18, dark-blue solid dots indicate that independent communication ranges of the *G2* robots in structural types Fa and Ra are twelve and eighteen unit areas, respectively. The blue and light-blue solid dots represent the distinct communication domains of the *G1* and *G3* automatons, encompassing sixteen and six units of area, correspondingly. Based on the above definitions, the independent communication ranges provided by six basic structural types are summarized in Table 8 (i.e., $\alpha_{1}$ to $\alpha_{6}$). However, if the left side of the focused *G2* robot corresponds to the structural type ${Ra}_{1}$, the focused *G2* robot’s two neighboring empty unit cells may be occupied by two *building* robots simultaneously (see Figure 19). In such cases, the focused *G2* robot will go dormant and stop broadcasting, causing the reduction of the total communication area. According to the *growing* robot’s working modes, the *growing* robot that doesn’t broadcast available empty unit cell coordinates is defined as the *dormant growing* sub-state (see Figure 2e). Although the overlapped communication area between neighboring *growing* robots can compensate for some of the losses, a *dormant G2* robot can still lead to a reduction in the communication area by four unit areas (see Figure 19).

Based on the definitions of the structural types, the calculation of the *dormant G2* robots is determined by Equation (30).
(30)$$D_{G2}=D_{G2}^{{Ra}_{1}}+D_{G2}^{{Ra}_{2}}+D_{G2}^{Fa}\;$$
where $D_{G2}^{R1}$, $D_{G2}^{R2}$, and $D_{G2}^{F}$ denote the number of *dormant G2* robots in structural types ${Ra}_{1}$, ${Ra}_{2}$, and Fa, respectively. The quantity of *dormant G2* robots within structural types ${Ra}_{1}$, i.e., $D_{G2}^{{Ra}_{1}}$, can be mathematically expressed as follows.
(31)$$D_{G2}^{{Ra}_{1}}=N_{{Ra}_{1}}\left(t\right)\tau^{*}\mu_{{Ra}_{1}}\mu_{{Ra}_{1}}$$
where the value of $\tau^{*}$ is determined by Equation (27). The $\mu_{{Ra}_{1}}$ is the overall probability of the *building* robot being assigned to the building unit cell provided by the *G2* robot of the structural type ${Ra}_{1}$. It is represented in Equation (32).
(32)$$\mu_{{Ra}_{1}}=\mu_{{Ra}_{1}}^{old}+\mu_{{Ra}_{1}}^{new}$$
here, $\mu_{{Ra}_{1}}^{old}$ and $\mu_{{Ra}_{1}}^{new}$ denote the overall probability that the building unit cells provided by the *G2* robot of structural type ${Ra}_{1}$ are occupied by *old* and *new building* robots, respectively. According to the structural features analysis in Equations (19)–(25), $\mu_{{Ra}_{1}}^{old}$ and $\mu_{{Ra}_{1}}^{new}$ can be written as follows.
(33)$$\mu_{{Ra}_{1}}^{old}=\frac{\xi \sigma_{old}+(N_{{Ra}_{1}}\left(t\right)-\xi )\epsilon_{old}}{N_{{Ra}_{1}}\left(t\right)+e_{0}}$$
(34)$$\mu_{{Ra}_{1}}^{new}=\frac{\xi \sigma_{new}+(N_{{Ra}_{1}}\left(t\right)-\xi )\epsilon_{new}}{N_{{Ra}_{1}}\left(t\right)+e_{0}}$$
where $e_{0}$ can be found in Equation (21). Similarly, the number of *dormant G2* robots within structural types ${Ra}_{2}$ and Fa, i.e., $D_{G2}^{{Ra}_{2}}$ and $D_{G2}^{Fa}$, can be written as follows, respectively.
(35)$$D_{G2}^{{Ra}_{2}}=N_{{Ra}_{2}}\left(t\right)\tau \mu_{{Ra}_{1}}\mu_{{Ra}_{2}}$$
(36)$$D_{G2}^{Fa}=N_{Fa}\left(t\right)\tau \mu_{{Ra}_{1}}\mu_{Fa}$$
where τ can be found in Equation (26). $\mu_{{Ra}_{2}}$ and $\mu_{Fa}$ represent the overall probability of the *building* robot distributing in the building unit cell provided by the *G2* robot in structural types ${Ra}_{2}$ and Fa, respectively. They are written as follows.
(37)$$\mu_{{Ra}_{2}}=\sigma_{new}+\sigma_{old}$$
(38)$$\mu_{Fa}=\mu_{Fa}^{old}+\mu_{Fa}^{new}$$
here, $\sigma_{old}$ and $\sigma_{new}$ are shown in Equations (21) and (24). The $\mu_{Fa}^{old}$ and $\mu_{Fa}^{new}$ are the overall probability of *old building* and *new building* robots distributing in the building unit cells provided by the *G2* robot in the structural type Fa. As analyzed in Equations (19)–(25), $\mu_{Fa}^{old}$ and $\mu_{Fa}^{new}$ can be expressed as
(39)$$\mu_{Fa}^{old}=\frac{(N_{{Fa}_{1^{*}}}\left(t\right)+N_{{Fa}_{2^{*}}}\left(t\right))\sigma_{old}+(N_{{Fa}_{1}}\left(t\right)+N_{{Fa}_{2}}\left(t\right))\epsilon_{old}}{N_{Fa}\left(t\right)+e_{0}}$$
(40)$$\mu_{Fa}^{new}=\frac{(N_{{Fa}_{1^{*}}}\left(t\right)+N_{{Fa}_{2^{*}}}\left(t\right))\sigma_{new}+(N_{{Fa}_{1}}\left(t\right)+N_{{Fa}_{2}}\left(t\right))\epsilon_{new}}{N_{Fa}\left(t\right)+e_{0}}$$

#### 3.3.2. The State Transition Function $\Gamma_{M}^{F}\left(t\right)$

The $\Gamma_{M}^{F}\left(t\right)$ signifies the count of *moving* robots transitioning into *free* robots when they fail in competition with others for the same target position. This state transition function can be expressed as Equation (41).
(41)$$\;\Gamma_{M}^{F}\left(t\right)=N_{M}\left(t\right)\gamma_{f}$$
where $\gamma_{f}$ is the probability of the *moving* robots failing to compete for the target position. Indeed, $\gamma_{f}$ is closely related to the working modes of the *growing* robots. Here, the *growing* robot that can provide an independent available empty unit cell for *moving* robots is regarded as being in the *working growing* sub-state, recorded as $G_{w}$ (see Figure 2e). According to the proposed self-assembly strategy, some available empty unit cell coordinates only impact the *free* robots, not the *moving* ones. If the provided available empty unit cells are shared or do not affect the *moving* robots, the *growing* robot is considered to be in the *activated growing* sub-state (see Figure 2e). Each available empty unit cell affecting *moving* robots is shared by $n_{0}$ *moving* robots, i.e.,
(42)$$n_{0}=\frac{N_{M}\left(t\right)}{N_{G_{w}}\left(t\right)}$$
where $N_{G_{w}}\left(t\right)$ is the number of *working growing* robots. Therefore, the success and failure rates of *moving* robots competing to obtain a target position are $\gamma_{s}$ and $\gamma_{f}$, i.e.,
(43)$$\gamma_{s}=\frac{1}{n_{0}}$$
(44)$$\gamma_{f}=1-\frac{1}{n_{0}}$$

The failure probability $\gamma_{f}$ relies on the count of working *growing* robots $N_{G_{w}}\left(t\right)$.

According to the structural features, the calculation of the *working growing* state numbers for *G1*, *G2*, and *G3* robots proceeds as follows. To determine *G1* robots in the *working growing* state, two probabilistically independent events should be discussed. The first event is the probability of the *G1* robot in *new growing* state. Here, the robot transformed from the *building* to the *growing* state at time step t−1 is defined as the *new growing* robot at time step t, or it is regarded as the *old growing* robot (see Figure 2d). Following the proposed self-assembly strategy, the broadcasts of the *new growing* robots exclusively impact the *free* robots and not the *moving* robots. Thus, all *new growing* robot is in *activated growing* state. The probability of the *G1* robot in a *new growing* state is expressed as $\rho_{G1}^{new}$, i.e.,
(45)$$\rho_{G1}^{new}=\frac{\Lambda_{B}^{G1}\left(t-1,T_{B}\right)}{N_{G1}\left(t\right)}$$
here, $\Lambda_{B}^{G1}$ is introduced in Section 3.3.4. The second event is the distribution characteristics of *building* robots within the building unit cells. Note that the possible building unit cell not occupied by a *building* robot will be preferentially selected as the available empty unit cell. Here, two situations should be considered. In the first situation, the building unit cell provided by a *G1* robot is not occupied by a *building* robot. Note that this building unit cell must be shared by the neighboring *G2* robot (see Section 3.2.3). Although the focused *G1* robot and its neighboring *G2* robot broadcast the building unit cell’s coordinates simultaneously, only the *G2* robot is usually considered the *working growing* state to avoid double counting. However, if the *G2* robot is in a *new growing* state, the focused *G1* robot could be considered the *working growing* robot. In addition, as *G1* robot has three neighboring empty unit cells. Thus, when its building unit cell is occupied, the focused *G1* robot will be in the *working growing* state. Thus, $N_{{G1}_{w}}^{F}\left(t\right)$ is expressed as follows.
(46)$$N_{{G1}_{w}}^{Fa}\left(t\right)=\left[N_{{Fa}_{1^{*}}}\left(t\right)+N_{{Fa}_{2^{*}}}\left(t\right)\right]\left(1-\rho_{G1}^{new}\right)[\left(1-\sigma_{old}-\sigma_{new}\right)\rho_{G2}^{new}+\sigma_{old}]$$
where $\left[N_{{Fa}_{1^{*}}}\left(t\right)+N_{{Fa}_{2^{*}}}\left(t\right)\right]$ expresses the total number of the *G1* robot in structural types ${Fa}_{1^{*}}$ and ${Fa}_{2^{*}}$. The second terms on the right-hand side of Equation (46) indicate the discussed first events. Two sub-terms in the third term represent the discussed two situations in the second events, respectively. $\rho_{G2}^{new}$ can be found in Equation (47).

Similarly, two probabilistically independent events need to be discussed to calculate the number of *G2* robots in the *working growing* state. The first event is the probability of the *G2* robots in a *new growing* state, i.e.,
(47)$$\rho_{G2}^{new}=\frac{\Lambda_{B}^{G2}\left(t-1,T_{B}\right)}{N_{G2}\left(t\right)}$$

The second event is the structural features involving the arrangement characteristics of the structural types and the distribution characteristics of *building* robots within different building unit cells (See Section 3.2). Through a comprehensive analysis of the structural features, the *G2* robot may be in the *working growing* state under the following three situations. The first situation is that the building unit cell provided by the focused *G2* robot is not occupied by a *building* robot. Under this situation, the focused *G2* robot is in the *working growing* state, providing an available empty unit cell for *free* and *moving* robots. The second situation is that an *old building* robot occupies the building unit cell provided by the focused *G2* robot, and the left structural type of the focused *G2* robot is not the structural type ${Ra}_{1}$. Note that the *G2* robot has two empty unit cell. According to the definition of the *building* robot’s distribution, the focused *G2* robot cannot actively use its second neighboring empty unit cell as a building unit cell (see Figure 14). Since the structural type on the left is not ${Ra}_{1}$, the focused *G2* robot can monopolize its second neighboring empty unit cell and work as a *working growing* robot to broadcast its second neighboring empty unit cell’s coordinates. In the third situation, an *old building* robot occupies the focused *G2* robot’s building unit cell, and the left structural type is ${Ra}_{1}$. As shown in Figure 16, the focused *G2* robot’s second neighboring empty unit cell plays the role of the building unit cell of the *G2* robot of the structural type ${Ra}_{1}$. When the building unit cell of the *G2* robot of the structural type ${Ra}_{1}$ is not occupied, the *G2* robot of the structural type ${Ra}_{1}$ is prioritized as the *working growing* robot rather than the focused *G2* robot to avoid double counting. However, if the *G2* robot of the left structural type ${Ra}_{1}$ is in the *new growing* state and its building unit cell is not occupied, the focused *G2* robot will be seen as a *working growing* robot. Additionally, a *new building* robot occupying a building unit cell indicates that the corresponding *G2* robot completed a recruitment task at time step t−1. Hence, the *G2* robot cannot function as a *working growing* robot, as its broadcasts can only impact the *free* robots at time step t. Based on the above discussions, the number of *G2* robots in structural type ${Ra}_{1}$ in the *working growing* state can be expressed as $N_{{G2}_{w}}^{{Ra}_{1}}(t)$, i.e.,
(48)$$N_{{G2}_{w}}^{{Ra}_{1}}\left(t\right)=N_{{Ra}_{1}}\left(t\right)\left(1-\rho_{G2}^{new}\right)[\left(1-\mu_{{Ra}_{1}}\right)+\mu_{{Ra}_{1}}^{old}\left(1-\tau^{*}\right)\;+\mu_{{Ra}_{1}}^{old}\tau^{*}\left(1-\mu_{{Ra}_{1}}\right)\rho_{G2}^{new}]$$
where $N_{{Ra}_{1}}\left(t\right)$ expresses the number of *G2* robots in the structural type ${Ra}_{1}$. Similarly, the number of *G2* robots in structural types ${Ra}_{2}$ and Fa in the *working growing* state are written as $N_{{G2}_{w}}^{{Ra}_{2}}\left(t\right)$ and $N_{{G2}_{w}}^{Fa}\left(t\right)$, i.e.,
(49)$$N_{{G2}_{w}}^{{Ra}_{2}}\left(t\right)=N_{{Ra}_{2}}\left(t\right)\left(1-\rho_{G2}^{new}\right)[\left(1-\mu_{{Ra}_{2}}\right)+\sigma_{old}\left(1-\tau \right)+\sigma_{old}\tau \left(1-\mu_{{Ra}_{1}}\right)\rho_{G2}^{new}]$$
(50)$$N_{{G2}_{w}}^{Fa}\left(t\right)=N_{Fa}\left(t\right)\left(1-\rho_{G2}^{new}\right)[\left(1-\mu_{Fa}\right)+\mu_{Fa}^{old}\left(1-\tau \right)+\mu_{Fa}^{old}\tau \left(1-\mu_{{Ra}_{1}}\right)\rho_{G2}^{new}]$$

As introduced in Section 3.2.3, the *G3* robots cannot enter a *new growing* state, and their only neighboring empty unit cells are the priority building unit cells by default. In the structural types ${Fa}_{2}$ and ${Fa}_{2^{*}}$, the *G3* robot can use its building unit cell preferentially. Thus, as long as a *building* robot does not occupy the building unit cell, the *G3* robot in the structural types ${Fa}_{2}$ and ${Fa}_{2^{*}}$ will remain in the *working growing* state. Based on the structural features, the quantity is expressed as $N_{{G3}_{w}}^{Fa}(t)$, i.e.,
(51)$$N_{{G3}_{w}}^{Fa}\left(t\right)=\left[N_{{Fa}_{2}}\left(t\right)+N_{{Fa}_{2^{*}}}\left(t\right)\right]\left(1-\sigma_{old}-\sigma_{new}\right)$$

In the structural type ${Ra}_{2}$, *G2* and *G3* robots share the building unit cell. When the *building* robot does not occupy the building unit cell, the *G2* robot is prioritized as the *working growing* robot to avoid double counting. However, if the *G2* robot is in the *new growing* state, the *G3* robot may be in the *working growing* state. It is expressed as follows.
(52)$$N_{{G3}_{w}}^{{Ra}_{2}}\left(t\right)=N_{{Ra}_{2}}\left(t\right)\left(1-\sigma_{old}-\sigma_{new}\right)\rho_{G2}^{new}$$

Finally, the total number of the *working growing* robots is written as follows.
(53)$$N_{G_{w}}\left(t\right)=N_{{G2}_{w}}^{{Ra}_{1}}\left(t\right)+N_{{G2}_{w}}^{{Ra}_{2}}\left(t\right)+N_{{G2}_{w}}^{Fa}\left(t\right)+N_{{G3}_{w}}^{Fa}\left(t\right)+N_{{G3}_{w}}^{{Ra}_{2}}\left(t\right)+N_{{G1}_{w}}^{Fa}\left(t\right)$$

#### 3.3.3. The State Transition Functions $\Gamma_{M}^{B}\left(t\right)$ and $\Lambda_{M}^{F}\left(t,T_{M}\right)$

According to the self-assembly strategy, all failed *moving* robots in competition will actively degrade to the *free* state. As shown in Equations (42) and (43), the number of *moving* robots winning the competition equals to the number of available empty unit cells, i.e., $N_{G_{w}}\left(t\right)$. Based on **Assumption 1**, the probability of these winning *moving* robots transforming into *building* robots at time step t  can be regarded as a constant-coefficient φ. Indeed, like the ξ in Equation (20), the value of the constant-coefficient φ is also closely related to the autonomous actions of robots and is affected by the building time parameter $T_{B}$. Thus, based on lots of simulation results, the value of φ under different building time parameters $T_{B}$ is shown in Table 7. Then, the state transition function from *moving* robots to *building* robots is written as in Equation (54), i.e.,
(54)$$\Gamma_{M}^{B}\left(t\right)=\varphi N_{G_{w}}\left(t\right)$$

The $\Lambda_{M}^{F}\left(t,T_{M}\right)$ represents the number of *moving* robots transitioning into *free* robots due to running out of the moving time parameter $T_{M}$. It is equivalent to the number of *moving* robots that have not changed their state during the time interval [$t-T_{M}$, t]. Referring to Ref. [16], $\Lambda_{M}^{F}\left(t,T_{M}\right)$ can be expressed as follows.
(55)$$\Lambda_{M}^{F}\left(t,T_{M}\right)=\Gamma_{F}\left(t-\tau_{m}\right)\prod_{k=t-\tau_{m}+1}^{t}\left(1-\gamma_{f}-r_{M}^{B}(t)\right)$$
where $\Gamma_{M}\left(t-\tau_{m}\right)$ represents the number of *moving* robots transformed from the *free* robots at time step $t-\tau_{m}$. $\gamma_{f}$ can be found in Equation (44). The $r_{M}^{B}(t)$ is the probability of *moving* robots transforming into a *building* state at time step t. It is expressed as follows.
(56)$$r_{M}^{B}\left(t\right)=\frac{\Gamma_{M}^{B}\left(t\right)}{N_{M}\left(t\right)}$$

#### 3.3.4. The State Transition Functions $\Lambda_{B}^{G}\left(t,T_{B}\right)$, $\Lambda_{B}^{G1}\left(t,T_{B}\right)$, and $\Lambda_{B}^{G2}\left(t,\tau_{B}\right)$

The state transition function $\Lambda_{B}^{G}(t,T_{B})$ represents the number of robots transitioning from *building* state to *growing* state at time step t. As introduced in Section 2.2, all *building* robots generated at the $t-T_{B}$ will become the *growing* robots at time step t. Thus, the $\Lambda_{B}^{G}(t,T_{B})$ is written as follows.
(57)$$\Lambda_{B}^{G}\left(t,T_{B}\right)=\Gamma_{M}^{B}\left(t-T_{B}\right)$$

As analyzed in Section 3.2.3, the *old building* robots are preferentially distributed in the priority building unit cells. Thus, the robots transitioning from *building* to *growing* state will be preferentially distributed in the priority building unit cells. Its probability can be written as follows.
(58)$$\sigma_{\Lambda_{B}}=\frac{\Lambda_{B}^{G}\left(t,T_{B}\right)}{N_{bu_{p}}\left(t\right)+e_{0}}$$

Correspondingly, the probability of these robots being distributed in the non-priority building unit cells is as follows.
(59)$$\epsilon_{\Lambda_{B}}=\frac{\Lambda_{B}^{G}\left(t,T_{B}\right)-\sigma_{\Lambda_{B}}N_{bu_{p}}\left(t\right)}{N_{bu}\left(t\right)-N_{bu_{p}}\left(t\right)+e_{0}}$$

As shown in Figure 14, only the structural types ${Fa}_{1}$ and ${Fa}_{2}$ can create *G1* robots. Thus, the number of robots transforming from *building* to *G1* state can be written as follows.
(60)$$\Lambda_{B}^{G1}\left(t,T_{B}\right)=N_{{Fa}_{1}}\left(t\right)\epsilon_{\Lambda_{B}}+N_{{Fa}_{2}}\left(t\right)[\sigma_{\Lambda_{B}}\left(1-\epsilon_{\Lambda_{B}}\right)+\epsilon_{\Lambda_{B}}\left(1-\sigma_{\Lambda_{B}}\right)]$$

Similarly, the *G2* robots can born in the structural types ${Ra}_{1}$, ${Ra}_{2}$, ${Fa}_{1^{*}}$, and ${Fa}_{2^{*}}$. Note that the structural type ${Fa}_{2}$ provides two adjacent building unit cells. Thus, when two *building* robots simultaneously occupy these two adjacent building unit cells, they will become the *G2* sub-state together. The number of robots transforming from *building* to *G2* state can be written as follows.
(61)$$\begin{array}{ll} \Lambda_{B}^{G2}\left(t,\tau_{B}\right)= & \sigma_{\Lambda_{B}}\left(N_{{Ra}_{2}}\left(t\right)+N_{{Fa}_{1^{*}}}\left(t\right)+2N_{{Fa}_{2^{*}}}\left(t\right)+\xi \right)+\epsilon_{\Lambda_{B}}\left(N_{{Ra}_{1}}\left(t\right)-\xi \right)\\ & \;\;+2N_{{Fa}_{2}}\left(t\right)\sigma_{\Lambda_{B}}\epsilon_{\Lambda_{B}} \end{array}$$

#### 3.3.5. The State Transition Functions ${∆}_{G1}^{G2}\left(t\right)$, ${∆}_{G1}^{G3}\left(t\right)$, ${∆}_{G2}^{G3}\left(t\right)$, and ${∆}_{G3}^{S}\left(t\right)$

As *building* robots transform into a *growing* state, the existing *growing* robots will upgrade their sub-states or transform into the *solid* state. In the structural types ${Fa}_{1^{*}}$ and ${Fa}_{2^{*}}$, creating a *growing* robot will make the *G1* robot update to the *G2* robot. Thus, the number of *G1* robots updating to the *G2* sub-state can be expressed in Equation (62) based on the analysis in Equations (58) and (59).
(62)$${∆}_{G1}^{G2}\left(t\right)=\sigma_{\Lambda_{B}}N_{{Fa}_{1^{*}}}\left(t\right)+2\sigma_{\Lambda_{B}}(1-\sigma_{\Lambda_{B}})N_{{Fa}_{2^{*}}}\left(t\right)-\Omega_{G1}^{*}\left(t\right)$$
here, $\Omega_{G1}^{*}\left(t\right)$ represents the number of *G1* robots that trigger the constraints of the shape boundary and become the *solid* robot directly during the update of the sub-state. The *G1* robots described by $\Omega_{G1}^{*}\left(t\right)$ belong to the *old growing* robots. However, it is also possible that a *growing* robot triggering shape boundary constraints belongs to the *new growing* robot. The new and old states of these *growing* robots depend on the specified 2D shape and the stochastic factors in the self-assembly process. Thus, for generality and simplicity, the $\Omega_{G1}^{*}\left(t\right)$ is defaulted to half of $\Omega_{G1}\left(t\right)$, i.e.,
(63)$$\Omega_{G1}^{*}\left(t\right)=0.5\Omega_{G1}\left(t\right)$$
where $\Omega_{G1}\left(t\right)$ can be found in Equation (72). As shown in Figure 14f, when two *growing* robots are born in these two building unit cells simultaneously, the *G1* robot in the structural type ${Fa}_{2^{*}}$ will directly update to the *G3* state. Thus, ${∆}_{G1}^{G3}\left(t\right)$ can be written as follows.
(64)$${∆}_{G1}^{G3}\left(t\right)={\sigma_{\Lambda_{B}}}^{2}N_{S_{{F2}^{*}}}\left(t\right)$$

Each structural type has a *G2* robot. Thus, the number of *G2* robots transforming into *G3* robots can be expressed as follows.
(65)$$\begin{array}{ll} {∆}_{G2}^{G3}\left(t\right)= & \sigma_{\Lambda_{B}}\left(N_{{Fa}_{1^{*}}}\left(t\right)+N_{{Fa}_{2^{*}}}\left(t\right)+N_{{Ra}_{2}}\left(t\right)+2\xi \;\right)\\ & \;\;\;+\epsilon_{\Lambda_{B}}\left(N_{{Fa}_{1}}\left(t\right)+N_{{Fa}_{2}}\left(t\right)+2\left(N_{{Fa}_{1}}\left(t\right)-\xi \right)\right)-\Omega_{G2}^{*}\left(t\right) \end{array}$$

Here, $\Omega_{G2}^{*}\left(t\right)$ represents the number of *old G2* robots that trigger the constraints of the shape boundary and become the *solid* robot directly during the update of the sub-state. Referring to Equation (63), it can be expressed as follows.
(66)$$\Omega_{G2}^{*}\left(t\right)=0.5\Omega_{G2}\left(t\right)$$
where $\Omega_{G2}\left(t\right)$ can be calculated in Equation (71). *G3* robots exist in the structural types ${Fa}_{2}$, ${Fa}_{2^{*}}$, and ${Ra}_{2}$. Thus, the number of *G3* robots transforming into *solid* robots can be expressed as follows.
(67)$${∆}_{G3}^{S}\left(t\right)=\sigma_{\Lambda_{B}}(N_{{Fa}_{2}}\left(t\right)+N_{{Fa}_{2^{*}}}\left(t\right)+N_{{Ra}_{2}}\left(t\right))$$

Note that *G3* robot will not trigger the shape boundary constraints.

### 3.4. The Constraints of the Shape Boundary

The shape of the formation is the collective goal of the entire swarm robotics system, specified and designed by the user. However, the shape boundary can partition the cross-arranged unit cells, causing it to be incomplete. Thus, when situated along the shape boundary, *G1* and *G2* robots transition directly to the *solid* state due to this incomplete unit cell. The predefined shape allows us to quantify how the shape boundary constraints affect the state transitions of the *growing* robots. The calculation results can serve as the initial conditions for the macroscopic model.

Using the example of a four-pointed star shape formation comprising 285 robots, 68 robots are affected by the shape boundary (see Figure 20b). To simplify our analysis, the robots containing two or three neighboring filled unit cells are considered the *G2* robots triggering the shape boundary constraints. Others are considered the *G1* robots triggering the shape boundary constraints. Thus, as shown the blue and red hollow circles in Figure 20b, 60 *G2* and 8 *G1* robots trigger the shape boundary constraints. Additionally, the number of *growing* robots affected by the shape boundary at time step  t is assumed to increase linearly as the total number of *growing* and *solid* robots increases. Here $x\left(t\right)$ is used to express the total number of *growing* and *solid* robots in the formed formation at time step t. It is written as follows.
(68)$$x\left(t\right)=N_{G}\left(t\right)+N_{S}\left(t\right)$$

The following equations can express the relationship between the number of *G2* robots affected by the boundary and the total number of *growing* and *solid* robots x.
(69)$$\int_{x_{G2}^{0}}^{x_{G2}^{1}}k_{G2}x+b_{G2}\;dx=n_{t}$$
(70)$$k_{G2}x_{G2}^{0}+b_{G2}=0$$
where $n_{t}$ is the total number of *G2* robots affected by the shape boundary. In this case, $n_{t}$ is equal to 60. $x_{G2}^{0}$ is the maximum number of *growing* and *solid* robots that can be accommodated in the formed formation before the first *G2* robot triggers the boundary constraints (see Figure 20a). After the last *G2* robot triggers the boundary constraints, the number of robots in the formed formation is represented by $x_{G2}^{1}$ (see Figure 20b). Here, the $x_{G2}^{0}$ and $x_{G2}^{1}$ are 137 and 277, respectively. By simultaneously solving Equations (69) and (70), we obtain $k_{G2}$ = 3/490, and $b_{G2}$ = −411/490. Therefore, the number of *G2* robots affected by constraints of the shape boundary at time step  t  can be expressed as follows.
(71)$$\Omega_{G2}\left(t\right)=\int_{N_{G}\left(t-1\right)+N_{S}\left(t-1\right)}^{N_{G}\left(t\right)+N_{S}\left(t\right)}k_{G2}x+b_{G2}dx,\;x_{G2}^{1}\geq x>x_{G2}^{0}$$

Similarly, the number of *G1* robots affected by constraints of the shape boundary at time step t  can be expressed as follows.
(72)$$\Omega_{G1}\left(t\right)=\int_{N_{G}\left(t-1\right)+N_{S}\left(t-1\right)}^{N_{G}\left(t\right)+N_{S}\left(t\right)}k_{G1}x+b_{G1}dx,\;x_{G1}^{1}\geq x>x_{G1}^{0}$$

Taking Figure 20b as an example, the $k_{G1}$, $b_{G1}$, $x_{G1}^{0}$, and $x_{G1}^{1}$ are equal to 0.25, −69.25, 277, and 285.

## 4. Verification and Discussion

### 4.1. The Verification of the Macroscopic Model

The prediction performance is the key indicator used to verify the proposed macroscopic model. In this section, we introduce the prediction error index ($P_{EI}$) to assess the model’s prediction accuracy. The $P_{EI}$ considers the prediction accuracy of *moving*, *building*, *growing*, and *solid* states. Based on the Euclidean distance and relative error, the prediction error index is defined as follows.
(73)$$P_{EI}=\frac{1}{T}\sum_{t=1}^{T}\frac{\left|N_{M}^{M}\left(t\right)-N_{M}^{E}\left(t\right)\right|+\left|N_{B}^{M}\left(t\right)-N_{B}^{E}\left(t\right)\right|+\left|N_{G}^{M}\left(t\right)-N_{G}^{E}\left(t\right)\right|+|N_{S}^{M}\left(t\right)-N_{S}^{E}\left(t\right)|}{n^{E}(t)}$$
where T is the total time of the self-assembly formation. $N_{M}^{M}$, $N_{B}^{M}$, $N_{G}^{M}$, and $N_{S}^{M}$ are the predicted number of *moving*, *building*, *growing*, and *solid* robots generated by the proposed macroscopic model. $N_{M}^{E}$, $N_{B}^{E}$, $N_{G}^{E}$, and $N_{S}^{E}$ are the average results of *moving*, *building*, *growing*, and *solid* robots in multiple experiments. The experimental results may come from simulations or real-robot experiments. $n^{E}(t)$ can be expressed as follows.
(74)$$n^{E}\left(t\right)=N_{M}^{E}\left(t\right)+N_{B}^{E}\left(t\right)+N_{G}^{E}\left(t\right)+N_{S}^{E}\left(t\right)\;$$

In this section, the four-pointed star shape formation is selected to verify the prediction performance of the proposed macroscopic model. The discussion of prediction performance in the small-scale self-assembly formation uses the scenario shown in Figure 10a. Here, 15 robots are used to self-assembly formation. The values of building time parameter $T_{B}$ and moving time parameter $T_{M}$ are set as one step and four steps, respectively. Here, both the real-robot experiments based on Waxberry robots and simulations based on embodied simulators are carried out. The real-robot experiment is repeated 10 times, and the simulations is repeated 100 times. With the same system parameters and specified scale, the prediction results of the whole self-assembly formation process are also calculated based on the proposed macroscopic model. As shown in Figure 21, the predictions based on the macroscopic model, the average results of 100 simulations and the average results of 10 real-robot experiments are represented by blue solid line, red dotted line, and green solid line. The black line with dots represents the experiment result in Figure 10a. Figure 21 indicates that the proposed macroscopic model can accurately describe each state’s change process. Compared with the average results of 10 real-robot experiments and the average results of 100 simulations, the proposed model’s prediction error index for this small-scale self-assembly formation is 14.17% and 12.20%, respectively, according to Equation (73). Note that the predicted result of the macroscopic model is not a reproduction of a single experiment. Instead, it directly feeds back the average results obtained in multiple experiments under specified scenarios. Although some errors exist between the experiment results of Figure 10a and the model’s predicted results, their dynamic trends are consistent.

Indeed, when the size of the specified shape formation is very small, the self-assembly formation process is easily affected by random factors. The characteristics of the prediction performance of the proposed model cannot be displayed clearly. Therefore, a large-scale formation consisting of 300 robots is also used to verify the proposed model. The building time parameter $T_{B}$ and moving time parameter $T_{M}$ are set as three steps and four steps, respectively. After 100 simulations, the comparison between the simulation’s average results and the model’s predictions is shown in Figure 22. The proposed model’s prediction error index for this large-scale self-assembly formation is only 4.05% according to Equation (73). As shown in Figure 22, the difference between the prediction and simulation results mainly occurs in the second half of the self-assembly formation process. This is caused by the constraints of the shape boundary being triggered. Indeed, the constraints of the shape boundary may create other complex structural types not considered in Figure 14. Meanwhile, the impact of boundary constraints on the quantities of six basic structural types is also ignored in Section 3.2.1. Therefore, the accuracy of the model’s prediction results will decrease after triggering the boundary constraints. When predicting the number of *G1* robots, the error is particularly evident (see Figure 22f). However, since the number of *G1* robots is generally very small, this does not significantly increase in the prediction error index $P_{EI}$.

To assess the adaptability of the proposed macroscopic model, the prediction error indexes are tested across various parameters using the four-pointed star and hexagonal crystal as formation shapes. The formation scale involves approximately 15 robots, 300 robots, 1500 robots, and 3500 robots, with the building time parameter $T_{B}$ set from 1 to 6 steps. Based on the analysis in Figure 21, the prediction error index of less than 20% is considered credible. As shown in Figure 23, the prediction error indexes of the proposed model show a similar trend in the two shapes. The prediction error indexes decrease with the scale of the formation in both shapes. These observations suggest that the proposed model demonstrates high adaptability across different shapes and different formation scales. Additionally, the proposed model proves effective in predicting the change process of self-assembly (i.e., $P_{EI}<20\%$) when the building time parameter $T_{B}$ is less than five steps. Notably, when the $T_{B}$ is set to three steps, the prediction error index reaches its lowest point. Especially when the formation scale is up to about 3500 robots, the $P_{EI}$ is less than 4% in both shapes. However, the model exhibits poor applicability when the building time parameter exceeds four steps. This limitation can be attributed to the distribution characteristics of *building* robots discussed in Section 3.2.3. As shown in Figure 14, the proposed model considers only part of neighboring empty unit cells as the building unit cells for *building* robots. Yet, with an increase in the building time parameter, all neighboring empty unit cells have the potential to become building unit cells and be occupied by *building* robots. Unfortunately, such scenarios are not accounted for in the proposed model, leading to a decrease in its prediction performance. The preceding analysis highlights that the proposed model shows excellent adaptability in predicting the self-assembly formation across diverse shapes, scales, and system parameters.

### 4.2. The Discussions

Compared to simulation and real-robot experiments, the macroscopic models can more efficiently display swarm behavior under varying influence factors. In this Section, we will discuss what factors affect the forming efficiency of the proposed self-assembly formation by using the four-pointed star shape formation consisting of about 300 robots as an example. Based on the proposed macroscopic model, the impact of the system parameters, i.e., building time parameter $T_{B}$ and moving time parameter $T_{M}$, should be first discussed. As shown in Figure 24, the total time of the self-assembly formation will increase with the building time parameter $T_{B}$, regardless of the formation’s scale. It is evident that the larger the formation scale, the more significant the impact of the building time parameter on the forming efficiency. In the formation consisting of about 3500 robots, reducing the building time parameter from 4 steps to 1 step resulted in a remarkable 28.5% reduction in the time required for formation. In the formation consisting of about 15 robots, a 16% reduction in total self-assembly time is observed when the building time parameter decreases from 4 steps to 1 step.

The moving time parameter $T_{M}$ is used to avoid the downtime of the self-assembly due to the fault of the *moving* robots. The effect of the moving time parameter on forming efficiency is another concern, as more *moving* robots will be forced to degenerate into *free* robots by reducing the moving time parameter. To analyze the impact of the moving time parameter on forming efficiency, the proposed model calculates the self-assembly process for moving time parameters from one step to four steps, respectively. The building time parameter is defaulted to three steps. Note that the state transition function from *moving* robots to *building* robots at time step t, i.e., $\Gamma_{M}^{B}\left(t\right)$, determines the forming efficiency. Thus, $\Gamma_{M}^{B}\left(t\right)$ is selected to analyze the impact of the moving time parameter on forming efficiency. As shown in Figure 25, the forming efficiency of the self-assembly formation is almost constant, regardless of the value of the moving time parameter. Especially when the moving time parameter is more than four steps, these curves of the state transition function $\Gamma_{M}^{B}\left(t\right)$ values almost overlap. That is, the moving time parameter $T_{M}$ does not affect the formation efficiency of the self-assembly formation. The reason for this counterintuitive result may come from the design of the proposed self-assembly formation strategy. As introduced in Section 2.2, each target position is shared by multiple *moving* robots, meaning high redundancy. Although some *moving* robots are limited by the moving time parameter being forced to degenerate into *free* robots, their target positions will still be finished by other *moving* robots.

The simulation tests for the parameters $T_{B}$ and $T_{M}$ prove that the formation efficiency of the proposed self-assembly formation exhibits monotonicity on the building time parameter $T_{B}$ and is independent of the moving time parameter $T_{M}$. Indeed, the building time parameter $T_{B}$ generally depends on the performance of a single robots. The faster the *building* robot finishes its assembly work, the less time there is for the self-assembly formation. However, the moving time parameter $T_{M}$ has nothing to do with the performance of a single robot. These results illustrate that the proposed distributed self-assembly formation strategy fully utilizes the performance of individual robots. The higher the individual performance (i.e., reducing building time), the higher the forming efficiency of the proposed self-assembly formation. This points out ways of optimizing the efficiency of the self-assembly formation. Meanwhile, the analysis of moving time parameters $T_{M}$ indicates that the proposed self-assembly formation has strong self-stability. Parameters unrelated to individual performance do not affect the swarm’s forming efficiency.

## 5. Conclusions

In this paper, a high-performance distributed self-assembly formation strategy is proposed, inspired by crystallization. To convert the macroscopic swarm behaviors to multiple simple collaborative tasks, five robot states, i.e., *free*, *moving*, *building*, *growing*, and *solid*, are defined by imitating crystallization. Consequently, robots in different states spontaneously formed a specified 2D shape formation with a lattice structure through local interactions. The feasibility of the proposed strategy is verified using 15 Waxberry robots to form the four-pointed star shape and hexagonal crystal shape. Further self-assembly formation tests in different conditions and larger scales were also conducted, facilitated by an embodied simulator. The fully distributed strategy and adaptation in different formation shapes verify the flexibility of the proposed self-assembly formation. The scalability of the strategy is confirmed through tests encompassing formations ranging from approximately 15 robots to 3500 robots. In addition, since all *growing* robots contribute to the expansion of the formation, the proposed self-assembly formation exhibits high forming efficiency. A rare and precious super-linear trend further demonstrates the high forming efficiency. Specifically, as the formation scale increases from about 15 to 3500 robots, the efficiencies of forming the four-pointed star shape and the hexagonal crystal shape formations are increased by 1756% and 2417%, respectively. These results collectively affirm the flexibility, scalability, and high efficiency of the proposed self-assembly formation strategy. The proposed high-performance self-assembly strategy for swarm robotics holds significant implications for advancing the practical applications of swarm robotics. For example, in flood relief efforts, many aquatic robots can rapidly self-assemble into emergency floating bridges based on the proposed self-assembly strategy. In intelligent warehouse systems, multiple transport robots can self-assemble into different-sized transport equipment using the proposed self-assembly strategy, enabling automatic adaptation to transport items of varying sizes.

In addition, a non-spatial macroscopic model is further developed to predict and analyze the swarm behavior. Following the stock and flow diagram generated by FSM, the master equations of this model are created. To solve the challenges of constructing the model’s state transition functions caused by the complex spatial structure, the SFE method is first proposed. This method opens the door to modeling the model for the swarm behavior affected by complex spatial structure. Here, we define six basic structural types and discuss their characteristics in detail. On this basis, all state transition functions that can explain the transition mechanisms are constructed. The simulations and experiments show that the proposed model has excellent prediction performance (i.e., $P_{EI}<20\%$) when the value of the building time parameter $T_{B}$ is less than five steps. The analysis based on the model demonstrates that it fully utilizes the performance of individual robots. A shorter building time consumed by a single robot means a higher forming efficiency. Additionally, the analysis of the moving time parameter $T_{M}$ verifies that the proposed self-assembly formation has strong self-stability. Parameters independent of individual performance do not affect the swarm’s forming efficiency.

Due to the lack of more Waxberry robots, the self-assembly formation has not been tested on larger-scale real robots. Producing more Waxberry robots and completing the test with larger-scale real robots is a focus of future research. Furthermore, although the proposed macroscopic model can adapt to parameter changes within a certain range, this adaptability is restricted to regions where the building time parameter $T_{B}$ is less than five steps. Therefore, further improving the model so that it can be adapted to all scenarios is another important research topic for the future.

## Figures and Tables

**Figure 1 sensors-24-03081-f001:**
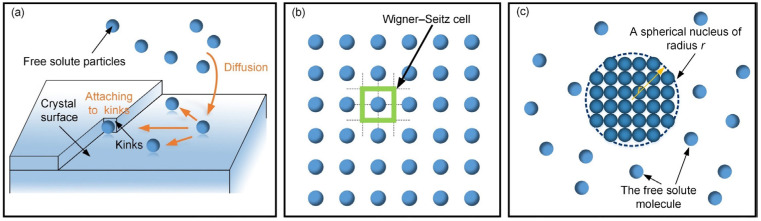
Three critical characteristics in crystallization. (**a**) The phase transition of the crystallization. Inspired by the phase transitions, the complex self-assembly formation task can be decomposed into several simple cooperative tasks among robots in different states. (**b**) The Wigner–Seitz unit cell in the parallelogram lattices. The unit cell is introduced for self-assembly to achieve the formation with lattice structures. (**c**) Nucleation. Nucleation in crystallization suggests that self-assembly formations also require a special robot to trigger this process.

**Figure 2 sensors-24-03081-f002:**
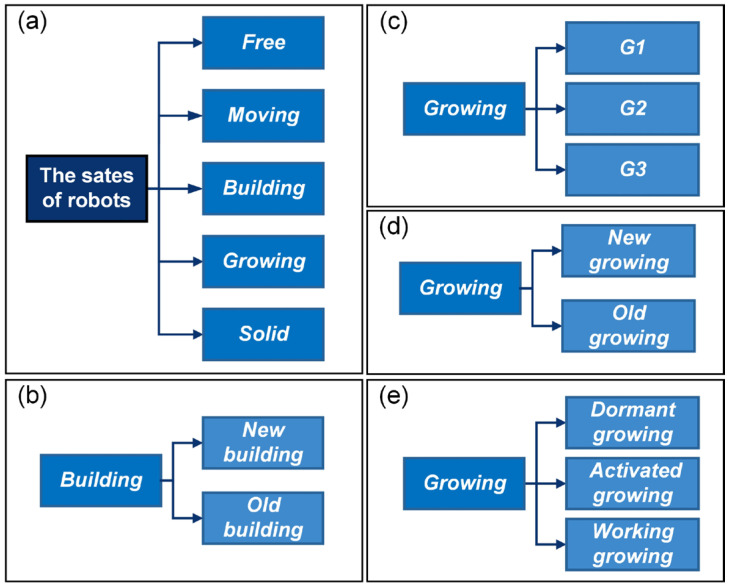
The definitions of robot’s states. (**a**) The main states of robots (**b**) The sub-states of *building* robots depend on birth time. (**c**) The sub-states of *growing* robots depend on the number of neighboring filled unit cells. (**d**) The sub-states of *growing* robots depend on birth time. (**e**) The sub-states of *growing* robots depend on working modes.

**Figure 3 sensors-24-03081-f003:**
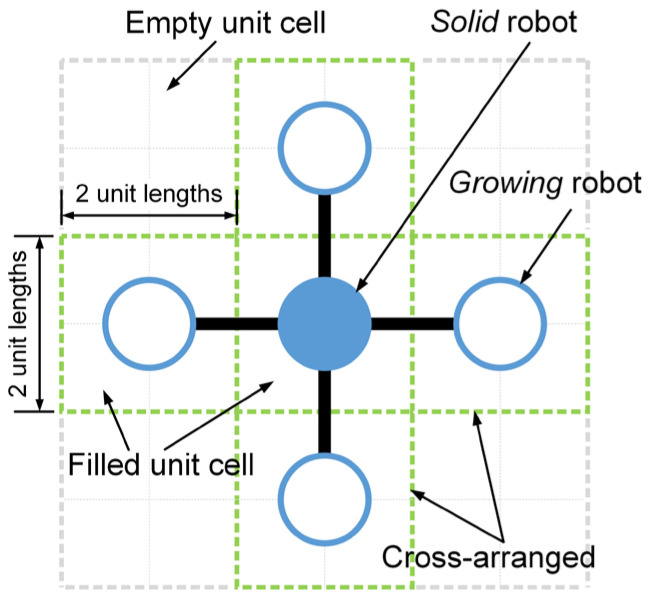
The cross-arranged unit cells for self-assembly formation inspired by Wigner–Seitz unit cell.

**Figure 4 sensors-24-03081-f004:**
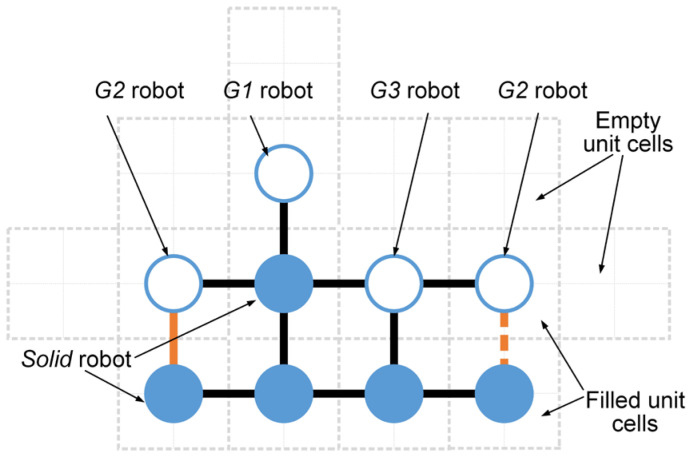
The sub-states of the *growing* state. Blue solid circle is the *Solid* robot. Blue hollow circle is the *growing* robot.

**Figure 5 sensors-24-03081-f005:**
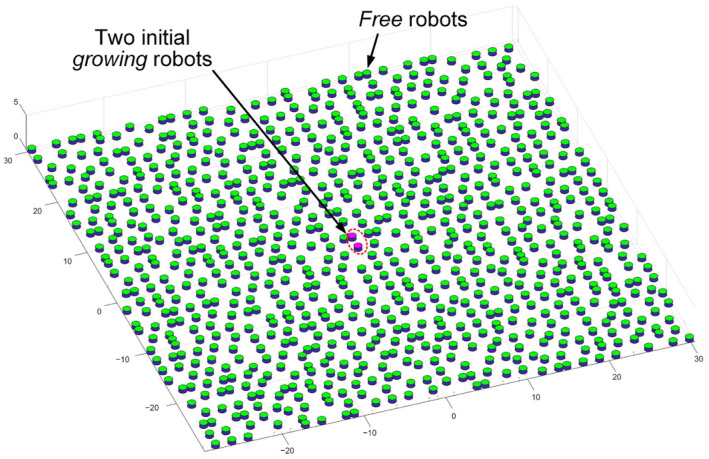
The initial conditions of the self-assembly formation and the specified initial *growing* robots inspired by nucleation. Two initial *growing* robots should be positioned at the coordinates $\left(x_{0},\;y_{0}\right)$ and $\left(x_{0},\;y_{0}+2\right)$. $x_{0}$ and $y_{0}$ can be assigned any value. Here, $\left(x_{0},\;y_{0}\right)$ is set as (0, 0).

**Figure 6 sensors-24-03081-f006:**
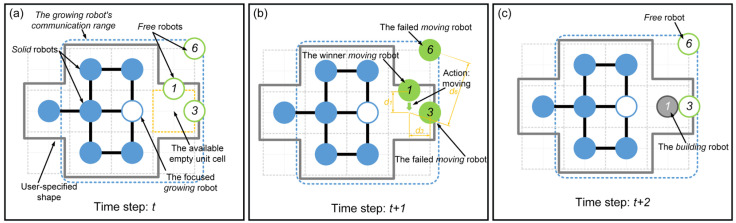
The competition mechanism of *moving* robots for the same target position. (**a**) Time step: t. (**b**) Time step: t+1. (**c**) Time step: t+2.

**Figure 7 sensors-24-03081-f007:**
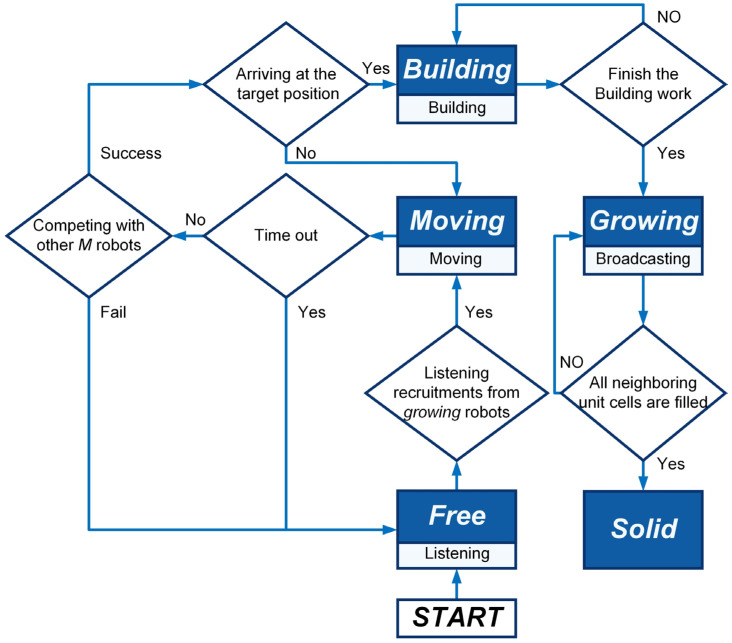
The FSM of a robot in the self-assembly formation.

**Figure 8 sensors-24-03081-f008:**
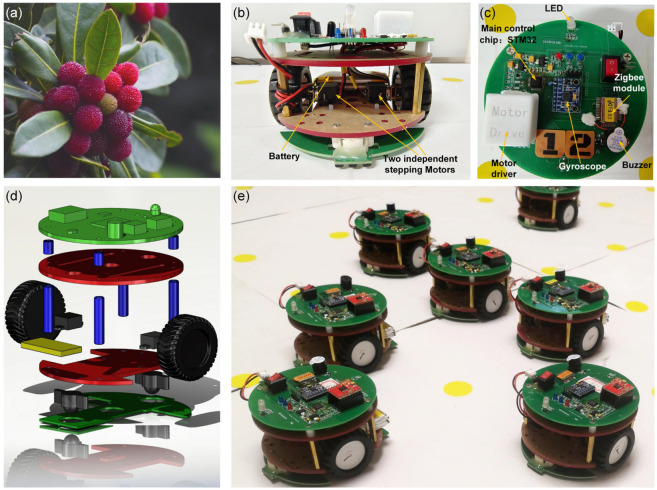
Waxberry and Waxberry robots. (**a**) Waxberry. (**b**) The battery and motors of the Waxberry robot. (**c**) The electronic system of the Waxberry robot. (**d**) The 3D design of the Waxberry robot. (**e**) The swarm robotics is composed of multiple Waxberry robots.

**Figure 9 sensors-24-03081-f009:**
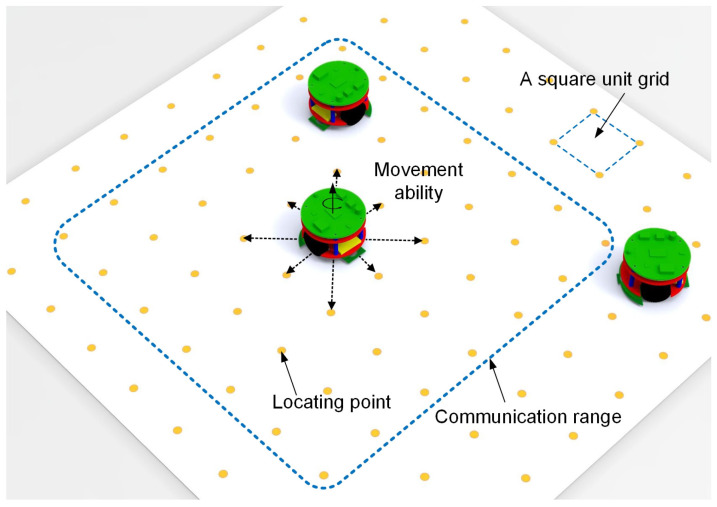
The foundational ability of the Waxberry robot.

**Figure 10 sensors-24-03081-f010:**
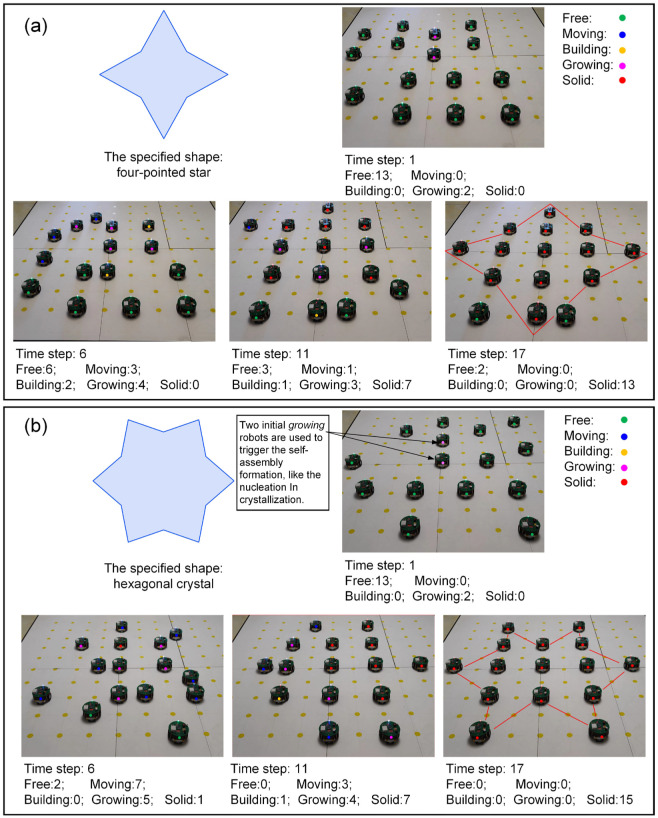
The self-assembly formation achieved by 15 Waxberry robots. (**a**) Self-assembly of a four-pointed star shape formation. (**b**) Self-assembly of a hexagonal crystal shape formation.

**Figure 11 sensors-24-03081-f011:**
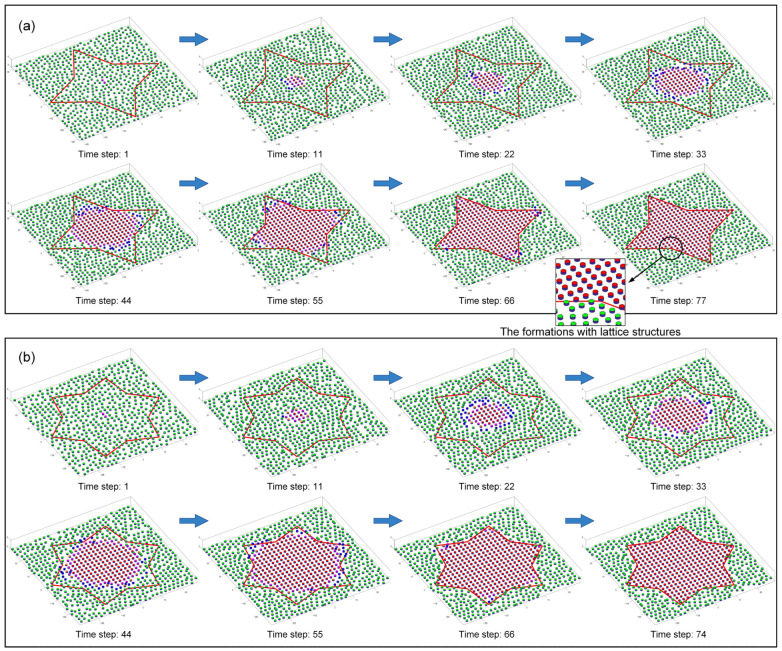
Self-assembly a formation comprising about 300 robots. (**a**) Self-assembly a four-pointed star shape formation. (**b**) Self-assembly a hexagonal crystal shape formation. Green dots: *free* robots. Blue dots: *moving* robots. Orange dots: *building* robots. Pink dots: *growing* robots. Red dots: *solid* robots.

**Figure 12 sensors-24-03081-f012:**
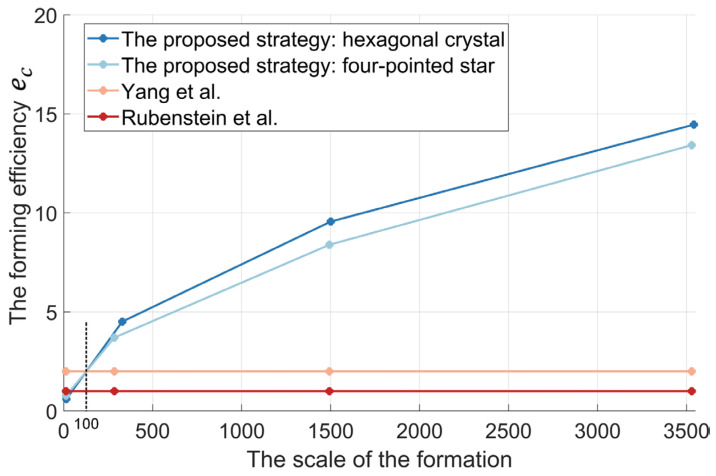
The super-linear of the proposed self-assembly formation based on simulation results [25,26].

**Figure 13 sensors-24-03081-f013:**
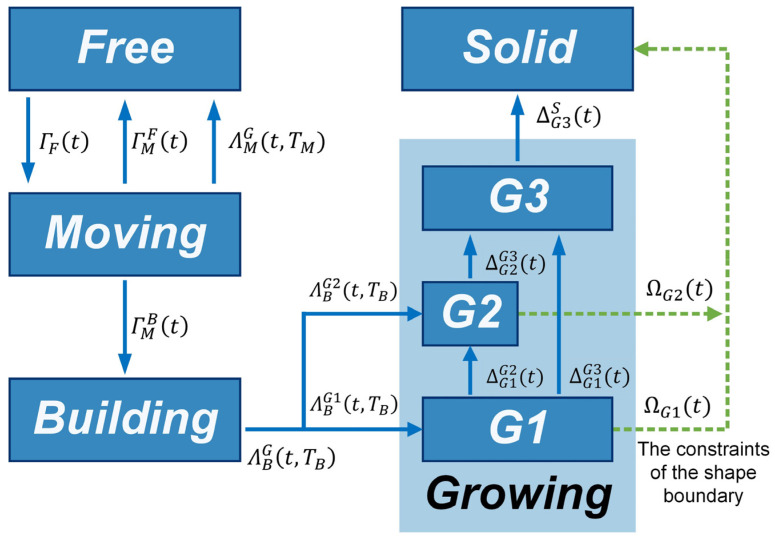
The stock and flow diagram of the proposed macroscopic model.

**Figure 14 sensors-24-03081-f014:**
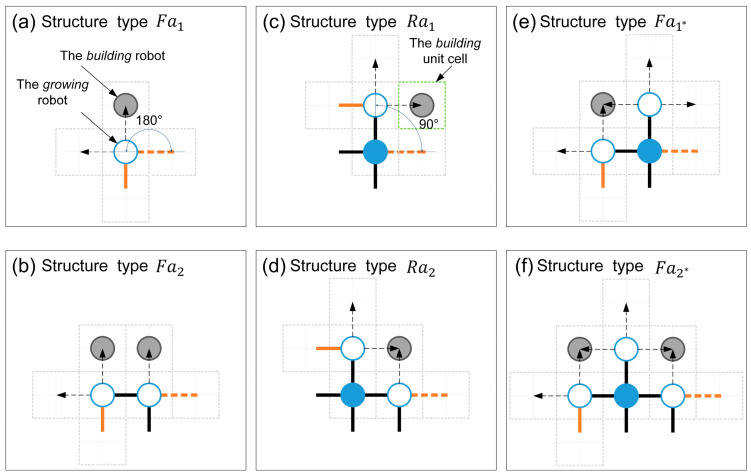
Six basic structural types. The orange heavy solid line and orange dotted line represent the connection interfaces to the previous and next structural types, respectively. The black dotted arrows points to the neighboring empty unit cells of the *growing* robots. The green dotted boxes represent the building unit cells, which are the empty unit cells allowed to be occupied by the *building* robots. Hollow blue circles represent *growing* robots, solid blue circles represent *solid* robots, and solid gray circles represent *building* robots.

**Figure 15 sensors-24-03081-f015:**
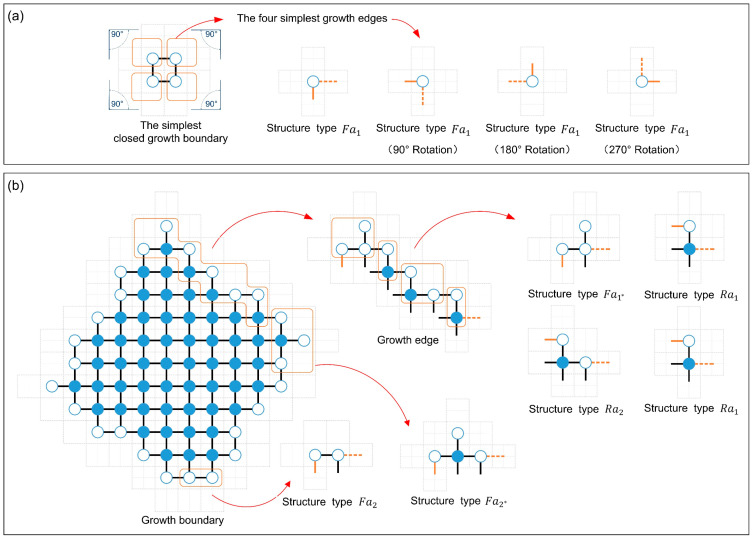
The relationship between growth boundary, growth edge, and structural types. (**a**) The disassembly of the simplest growth boundary. (**b**) The disassembly of a normal growth boundary.

**Figure 16 sensors-24-03081-f016:**
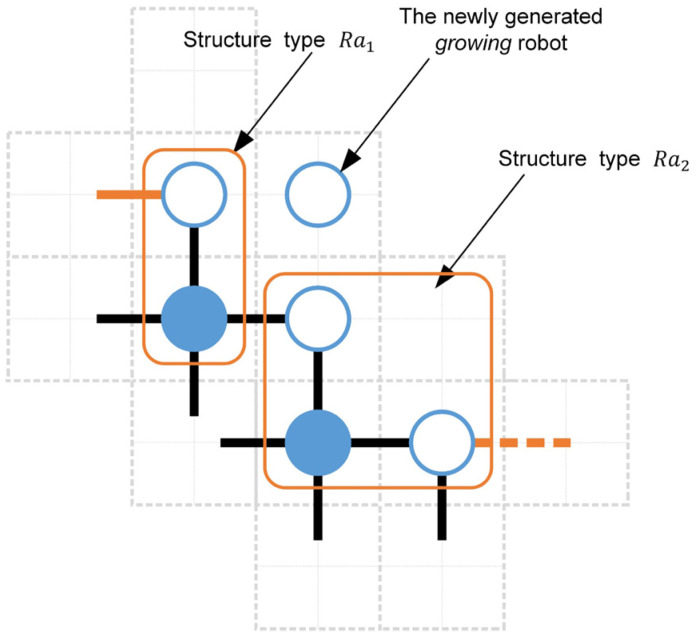
The effect of the structural type ${Ra}_{1}$ for other structural types.

**Figure 17 sensors-24-03081-f017:**
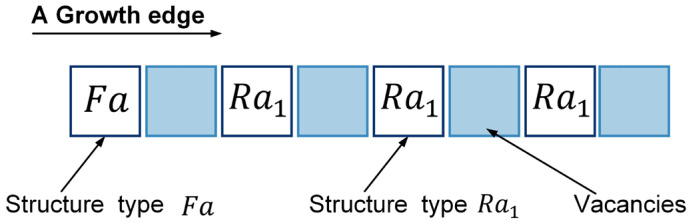
The arrangement approach of structural types within a growth edge.

**Figure 18 sensors-24-03081-f018:**
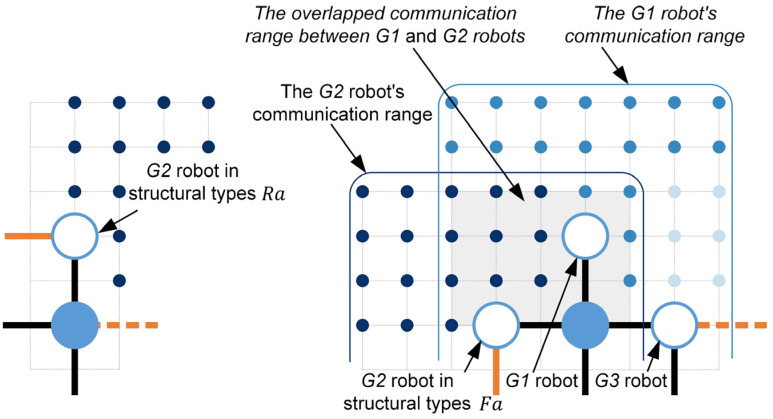
The definition of the independent communication ranges of *growing* robots. Blue solid dots: *G1* robot’s independent communication ranges. Dark blue solid dots: *G2* robot’s independent communication ranges. Light blue solid dots: *G3* robot’s independent communication ranges.

**Figure 19 sensors-24-03081-f019:**
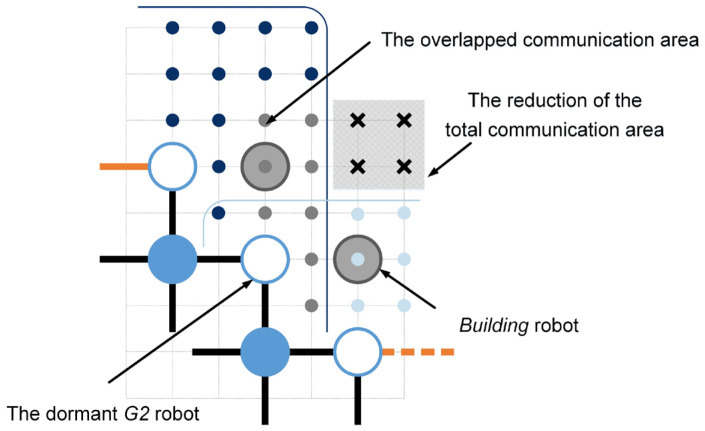
The effects of the *dormant G2* robot for total communication area. Gray solid dots: the portion of the communication ranges overlap between a *G2* robot and its neighboring *growing* robots. Black fork: The reduction of the total communication area.

**Figure 20 sensors-24-03081-f020:**
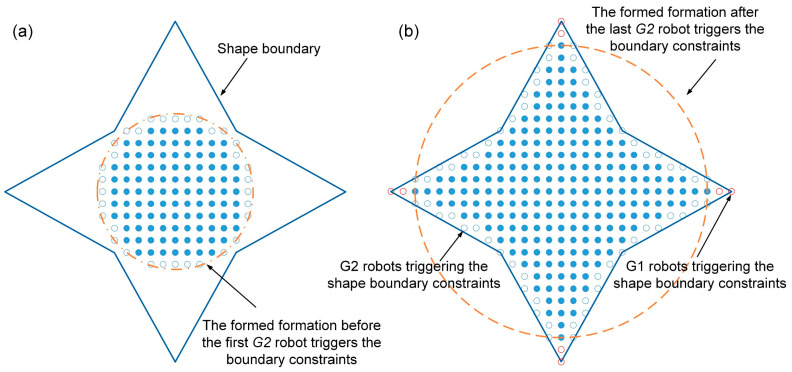
The calculating example of the constraints of the shape boundary. (**a**) The initial condition triggers the constraints of the shape boundary. (**b**) The finial condition triggers the constraints of the shape boundary.

**Figure 21 sensors-24-03081-f021:**
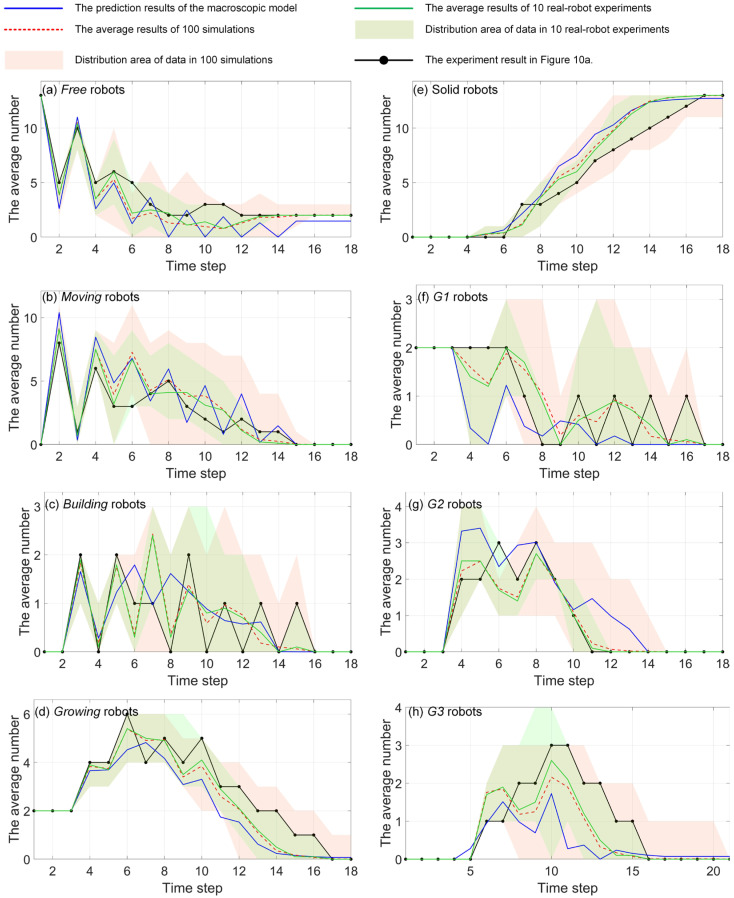
The analysis of the prediction performance of the macroscopic model in a small-scale four-pointed star shape formation. Blue solid line: the prediction results of the macroscopic model. Red dotted line: the average results of 100 simulations. Red shadow: distribution area of data in 100 simulations. Green solid line: the average results of 10 real-robot experiments. Green shadow: distribution area of data in 10 real-robot experiments. The black line with dots: the experiment result in Figure 10a.

**Figure 22 sensors-24-03081-f022:**
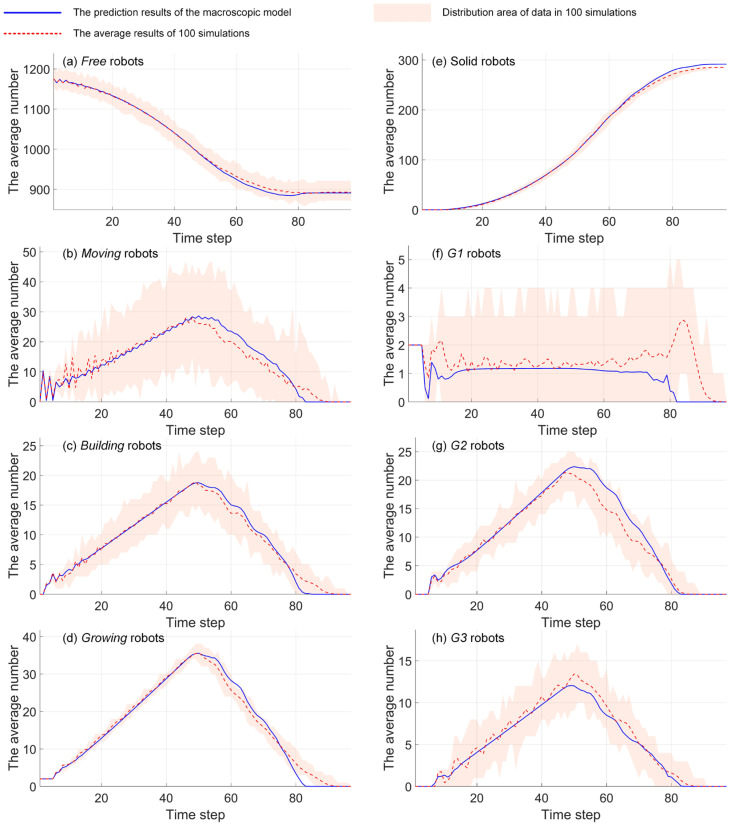
The analysis of the prediction performance of the macroscopic model in a large-scale four-pointed star shape formation. Blue solid line: the prediction results of the macroscopic model. Red dotted line: the average results of 100 simulations. Red shadow: distribution area of data in 100 simulations.

**Figure 23 sensors-24-03081-f023:**
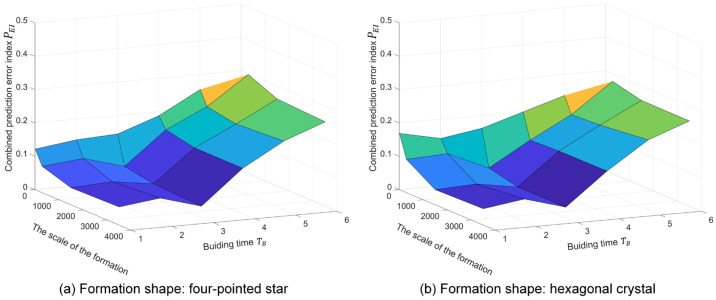
The prediction error indexes of the model in different conditions.

**Figure 24 sensors-24-03081-f024:**
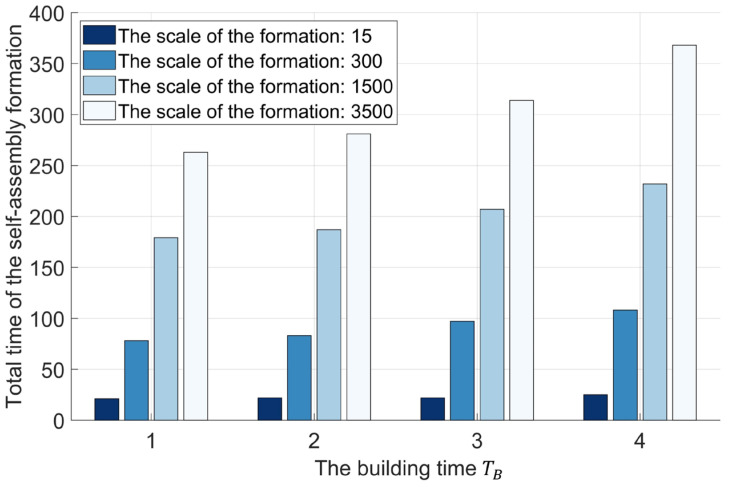
The effects of the building time parameter $T_{B}$ on the forming efficiency.

**Figure 25 sensors-24-03081-f025:**
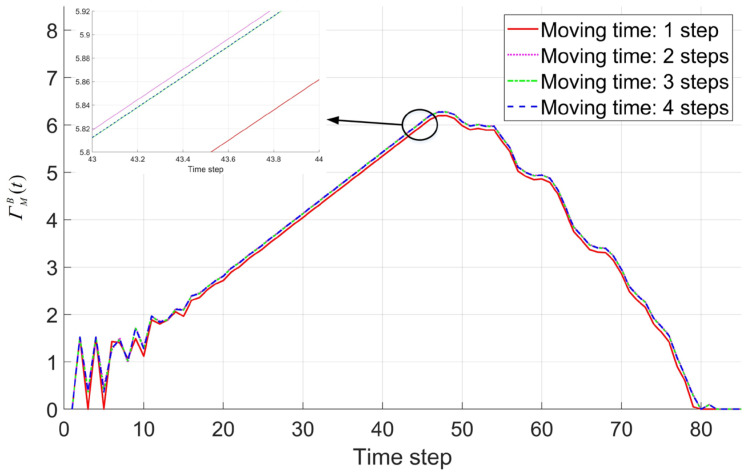
The effects of the moving time parameter $T_{M}$ on the forming efficiency.

**Table 1 sensors-24-03081-t001:** An overview of research on self-assembly of swarm robotics.

Reference	Authors	Research Achievements	Methods	Limitations
Ref. [25]	Rubenstein et al.	Swarm robotics organized a single motion chain to achieve self-assembly formation.	The FSM method	Extremely low efficiency resulted from only single-threaded mode.
Refs. [26,27]	Yang et al.	Swarm robotics organized two parallel motion chains to achieve self-assembly formation.	The FSM method	Low efficiency resulted from only double-threaded mode.
Ref. [28]	Divband Soorati et al.	Swarm robotics self-assembled to the tree-like formation to search for bright areas.	The FSM method	The constraints between father and child nodes limit efficiency.
Ref. [29]	Zhu et al.	Swarm robotics self-assembled to a square formation.	The FSM method	The robustness is limited due to designing a global leader for swarm.
Ref. [30]	Zheng et al.	Swarm robotics self-assembled to the specified formation.	The density-feedback method	The flexibility is limited due to rely on the off-line global pattern planning.
Ref. [31]	Deshmukh et al.	Swarm robotics self-assembled to the specified 2D formation.	The density-feedback method	The robustness is limited due to rely on a centralized controller.
Ref. [32]	Klavins et al.	Swarm robotics achieved small-scale self-assembly formation.	The graph-based method	Not suitable for large scale self-assembly formation.
Ref. [33]	Mong-ying et al.	Swarm robotics achieved small-scale self-assembly formation.	The graph-based method	Not suitable for large scale self-assembly formation.
Ref. [34]	Groβ et al.	Swarm robotics self-assembled into a small formation for collective transport.	The evolutionary algorithms	Not suitable for large scale self-assembly.
Ref. [35]	Sperati et al.	Swarm robotics self-assembled into a small-scale path.	The evolutionary algorithms	Not suitable for large scale self-assembly.
Ref. [36]	Khaldi et al.	Swarm robotics self-assembled to the specified path.	The potential field methods	Unable to construct a formation with lattice structures.
Ref. [37]	Cheah et al.	Swarm robotics self-assembled to the specified 2D formation.	The potential field methods	Unable to construct a formation with lattice structures.

**Table 2 sensors-24-03081-t002:** The definition of states of robot inspired by crystal growth.

The Phenomena of Crystal Growth	The States of Robot
The free solute particles	*Free* robots
The solute particles in the diffusion process	*Moving* robots
The solute particles entering kinks	*Building* robots
The atoms making up the crystal surface	*Growing* robots
The atoms inside crystal	*Solid* robots

**Table 3 sensors-24-03081-t003:** Classifications and definitions of the unit cells.

Definition of Unit Cells	Descriptions
Filled unit cell	The unit cells are filled by *growing* or *solid* robots.
Empty unit cell	The unit cells are not filled by *growing* or *solid* robots.
Available empty unit cell	The empty unit cells selected by *growing* robots and are allowed as target positions for *free* and *moving* robots.
Building unit cell	The empty unit cells are allowed to be occupied by *building* robots.
Priority building unit cell	The building unit cells can be occupied by *building* robots preferentially.
Non-priority building unit cell	The building unit cells are occupied by *building* robots non-preferentially.

**Table 4 sensors-24-03081-t004:** Description of notations in the master difference equations.

Notations	Descriptions
$N_{F}\left(t\right)$	The number of *free* robots at time step t.
$N_{M}\left(t\right)$	The number of *moving* robots at time step t.
$N_{B}\left(t\right)$	The number of *building* robots at time step t.
$N_{G}\left(t\right)$	The number of *growing* robots at time step t.
$N_{G1}\left(t\right)$	The number of *G1* robots at time step t.
$N_{G2}\left(t\right)$	The number of *G2* robots at time step t.
$N_{G3}\left(t\right)$	The number of *G3* robots at time step t.
$N_{S}\left(t\right)$	The number of *solid* robots at time step t.
$\Gamma_{F}\left(t\right)$	The number of *free* robots becoming *moving* robots at time step t due to receiving broadcasts.
$\Gamma_{M}^{F}\left(t\right)$	The number of *moving* robots becoming *free* robots at time step t due to failure in competition.
$\Lambda_{M}^{F}(t,T_{M})$	The number of *moving* robots becoming *free* robots at time step t due to running out of the moving time parameter $T_{M}$.
$\Gamma_{M}^{B}\left(t\right)$	The number of *moving* robots becoming *building* robots as they succeed in competition and arrive at their target positions at time step t.
$\Lambda_{B}^{G}(t,T_{B})$	The number of *building* robots becoming *growing* robots at time step t $\mathrm{after}\;\mathrm{spending}\;T_{B}$ steps assembling the formation.
$\Lambda_{B}^{G1}\left(t,T_{B}\right)$	The number of *building* robots becoming *G1* robots at time step t $\mathrm{after}\;\mathrm{spending}\;T_{B}$ steps assembling the formation.
$\Lambda_{B}^{G2}(t,T_{B})$	The number of *building* robots becoming *G2* robots at time step t $\mathrm{after}\;\mathrm{spending}\;T_{B}$ steps assembling the formation.
${∆}_{G1}^{G2}(t)$	The number of *G1* robots becoming *G2* robots at time step t.
${∆}_{G1}^{G3}(t)$	The number of *G1* robots becoming *G3* robots at time step t.
${∆}_{G2}^{G3}(t)$	The number of *G2* robots becoming *G3* robots at time step t.
${∆}_{G3}^{S}(t)$	The number of *G3* robots becoming *solid* robots at time step t.
$\Omega_{G1}(t)$	The number of *G1* robots becoming *solid* robots at time step t due to constraints of shape boundary.
$\Omega_{G2}(t)$	The number of *G2* robots becoming *solid* robots at time step t due to constraints of shape boundary.

**Table 5 sensors-24-03081-t005:** The comparison among the existing methods and our proposed method.

	Applicable to Spatial Structure Scenarios	Explaining the Inner Mechanisms of the State Transitions
Data statistics method [50,51,52,53]	√	×
Geometrical estimation method [14,16]	×	√
Structural feature estimation method	√	√

**Table 6 sensors-24-03081-t006:** Description of notations in all state transition functions.

Notations	Descriptions
$N_{Fa}(t)$	The number of structural types Fa.
$N_{{Fa}_{1}}\left(t\right)$	The number of structural types ${Fa}_{1}$.
$N_{{Fa}_{1^{*}}}\left(t\right)$	The number of structural types ${Fa}_{1^{*}}$.
$N_{{Fa}_{2}}\left(t\right)$	The number of structural types ${Fa}_{2}$
$N_{{Fa}_{2^{*}}}\left(t\right)$	The number of structural types ${Fa}_{2^{*}}$.
$N_{Ra}(t)$	The number of structural types Ra.
$N_{{Ra}_{1}}(t)$	The number of structural types ${Ra}_{1}$.
$N_{{Ra}_{2}}(t)$	The number of structural types ${Ra}_{2}$.
$N_{bu}\left(t\right)$	The total number of building unit cells.
$N_{bu_{p}}\left(t\right)$	The total number of the priority building unit cells.
$N_{B_{new}}\left(t\right)$	The number of *new building* robots.
$N_{B_{old}}\left(t\right)$	The number of *old building* robots.
$\sigma_{old}$	The probability of *old building* robots in the priority building unit cell.
$\epsilon_{old}$	The probability of *old building* robots in the non-priority building unit cell.
$\sigma_{new}$	The probability of *new building* robots in the priority building unit cell.
$\epsilon_{new}$	The probability of *new building* robots in the non-priority building unit cell.
τ	The probability of structural type ${Ra}_{1}$ being on the left side of structural types ${Ra}_{2}$ and Fa.
$\tau^{*}$	The probability of structural type ${Ra}_{1}$ being on the left side of structural type ${Ra}_{1}$.
$\mu_{{Ra}_{1}}$	The probability of the *building* robots in the building unit cell provided by the *G2* robots $\;\mathrm{of}\;\mathrm{the}\;\mathrm{structural}\;\mathrm{type}\;{Ra}_{1}$.
$\mu_{{Ra}_{1}}^{old}$	The probability of the *old building* robots in the building unit cell provided by the *G2* robots $\;\mathrm{of}\;\mathrm{the}\;\mathrm{structural}\;\mathrm{type}\;{Ra}_{1}$.
$\mu_{{Ra}_{1}}^{new}$	The probability of the *new building* robots in the building unit cell provided by the *G2* robots $\;\mathrm{of}\;\mathrm{the}\;\mathrm{structural}\;\mathrm{type}\;{Ra}_{1}$.
$\mu_{{Ra}_{2}}$	The probability of the *building* robots in the building unit cell provided by the *G2* robots $\;\mathrm{of}\;\mathrm{the}\;\mathrm{structural}\;\mathrm{type}\;{Ra}_{2}$.
$\mu_{Fa}$	The probability of the *building* robots in the building unit cell provided by the *G2* robots of the structural type Fa.
$\mu_{Fa}^{old}$	The probability of the *old building* robots in the building unit cell provided by the *G2* robots of the structural type Fa.
$\mu_{Fa}^{new}$	The probability of the *new building* robots in the building unit cell provided by the *G2* robots of the structural type Fa.
$\rho_{G1}^{new}$	The probability of the *G1* robots in a *new growing* state.
$\rho_{G2}^{new}$	The probability of the *G2* robots in a *new growing* state.
$\sigma_{\Lambda_{B}}$	The probability of the robot transitioning from *building* to *growing* state within the priority building unit cells.
$\epsilon_{\Lambda_{B}}$	The probability of the robot transitioning from *building* to *growing* state within the non-priority building unit cells.

**Table 7 sensors-24-03081-t007:** The value of coefficients ξ and φ.

$\mathrm{Building}\;\mathrm{Time}\;\mathrm{Parameter}\;T_{B}$ (Unit: Steps)	1	2	3	4	≥5
ξ	1.3	1.1	0.7	0.5	0.2
φ	0.62	0.74	0.76	0.78	0.8

ξ can be found in Equation (20). φ can be found in Equation (54).

**Table 8 sensors-24-03081-t008:** The value of coefficients $a_{0}$ to $a_{6}$.

$\alpha_{0}$	$\alpha_{1}$	$\alpha_{2}$	$\alpha_{3}$	$\alpha_{4}$	$\alpha_{5}$	$\alpha_{6}$
4	18	36	24	42	12	18

## Data Availability

Data are contained within the article.

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
