# Peer review of "Crystallization-Inspired Design and Modeling of Self-Assembly Lattice-Formation Swarm Robotics"

_sensors, 2024, doi:10.3390/s24103081_

Round 1

Reviewer 1 Report

Comments and Suggestions for Authors

1. Academically, self-assembly is the spontaneous organization of individual components into an ordered structure without external intervention. That means, self-assembly is highly decentralized. Implicitly, self-assembly is a bottom-up emergence of large-scale population interaction (Figure 11 can see this). However, the authors stated that 15 Waxberry robots they only used could form a hexagonal crystal shape formation. Although the authors further provided simulation-based self-assembly, the simulation seem different with the real swarm robots, for example, the growing mechanism. Even the mechanism of material crystallization is not imitated by so simple information interactions.

2. As for the macroscopic model, it is highly probabilistic, which means a possibility, not a necessity. If possible, hope the authors use the macroscopic model to prove the case in the Figure 9.

3. In the Figure 2, the specific meanings of G1, G2 and G3 are not indicated, and whether there is a corresponding relationship with Figure 3 should be explained in detail.

4. The so-called Waxberry robot is a common mobile car with two driving wheels. Please give the detail configuration of the robot to demonstrate it has the capability of omni-directional movement.

5. Please check the Assumptions in line 473 and 475, and reveal their substantial relations with the crystallization mechanism. 

Comments on the Quality of English Language

General

Reviewer 2 Report

Comments and Suggestions for Authors

Review

The paper investigates the crystallization inspired design and modeling of self-assembly lattice formation swarm robotics. After the introduction the paper discusses the self-assembly formation inspired by crystallization. After that, section 3 presents the mathematical description of the macroscopic model. Section 4 investigates the verification and discussion. The last section is the conclusion and future research direction. I have the following questions:

  • Literature review: please add a table which summarizes the literature review, for example with the following headers: reference, authors, problem, solving methods.

  • Table 6: Why these parameter values were applied?

  • In which programming language were the test runs implemented? The description of the program is missing.

  • Please highlight the practical applicability of the research.

Round 2

Reviewer 1 Report

Comments and Suggestions for Authors

The author revised the paper according to the review comments. I agree to accept its publication in its current form.